# AGENTMIXER: MULTI-AGENT CORRELATED POLICY FACTORIZATION

## ABSTRACT

Centralized training with decentralized execution (CTDE) is widely employed to stabilize partially observable multi-agent reinforcement learning (MARL) by utilizing a centralized value function during training. However, existing methods typically assume that agents make decisions based on their local observations independently, which may not lead to a correlated joint policy with sufficient coordination. Inspired by the concept of correlated equilibrium, we propose to introduce a *strategy modification* to provide a mechanism for agents to correlate their policies. Specifically, we present a novel framework, AgentMixer, which constructs the joint fully observable policy as a non-linear combination of individual partially observable policies. To enable decentralized execution, one can derive individual policies by imitating the joint policy. Unfortunately, such imitation learning can lead to *asymmetric learning failure* caused by the mismatch between joint policy and individual policy information. To mitigate this issue, we jointly train the joint policy and individual policies and introduce *Individual-Global-Consistency* to guarantee mode consistency between the centralized and decentralized policies. We then theoretically prove that AgentMixer converges to an $\epsilon$-approximate Correlated Equilibrium. The strong experimental performance on three MARL benchmarks demonstrates the effectiveness of our method.

## 1 INTRODUCTION

Cooperative multi-agent reinforcement learning (MARL) has attracted substantial attention in recent years owing to its promise in solving many real-world tasks that naturally comprise multiple decision-makers interacting at the same time, such as multi-robot control (Gu et al., 2023), traffic signal control (Ma & Wu, 2020), and autonomous driving (Shalev-Shwartz et al., 2016). However, unlike the single-agent RL settings, learning in multi-agent systems (MAS) poses two primary challenges: coordination, i.e., agents should work together in order to achieve a common goal and learn optimal joint behavior, and partial observability, which limits each agent to her own local observations and actions. To address these difficulties, most works adopt a popular learning framework called Centralized Training Decentralized Execution (CTDE) (Lowe et al., 2017; Yu et al., 2022; Rashid et al., 2020a) that allows agents to access global information during the training phase while remaining the learned policies executed with only local information in a decentralized way.

To enhance coordination, one line of research is to use value decomposition (VD) (Sunehag et al., 2017), e.g. QMIX (Rashid et al., 2020b) and QPLEX (Wang et al., 2021a), which learns a centralized joint action value function factorized by decentralized agent utility functions. With the structural constraint of Individual-Global-Max (IGM) (Son et al., 2019), it guarantees the optimal action consistency between the centralized and decentralized policies. On the other hand, multi-agent policy gradient (MAPG) methods(de Witt et al., 2020), such as MADDPG (Lowe et al., 2017) and MAPPO (Yu et al., 2022), has achieved remarkable success. However, while learning a centralized critic, previous works are still constrained by assuming independence among agents during exploration. A few recent works further propose auto-regressive policies to impose coordination among agents by allowing agents to observe other agents' actions, either explicitly (Fu et al., 2022; Wang et al., 2023) or implicitly (Li et al., 2023; Wen et al., 2022). Inspired by the Correlated Equilibrium (CE) (Maschler et al., 2013) in game theory, MAVEN (Mahajan et al., 2019) and SIC (Chen et al., 2022) introduce a hierarchical control method with an external shared latent variable as additional information for agents to coordinate each other. However, note that existing auto-regressive meth-

ods assume a pre-defined execution order. Moreover, most existing correlated policies violate the requirement for decentralized execution. This paper instead aims to achieve Correlated Equilibrium in a fully decentralized way which is crucial for real-world applications.

In order to mitigate the difficulty of learning under partial observability, CTDE exploits true state information, usually via a centralized critic, to train individual policies conditioned on the local observation-action history. While it is possible to first learn a centralized expert policy and then train the decentralized agents to follow it (Lin et al., 2022), it may result in suboptimal partially observable policies since the omniscient critic or agent has no knowledge of what the decentralized agents do not know, referred to as the *asymmetric learning failure* (Warrington et al., 2021). Consider a scenario where two agents of distinct physical shapes try to get to their opposite destinations through two possible paths 1 and 2, as shown in Figure 1. Successful policies should avoid collision as the body sizes of agents always change and each passage only permits two small agents or one big agent. In CTDE, we can learn the optimal fully observable joint policy conditioned on agents' physiques, which would select the shorter path 1 when both agents are small. However, naively learn-

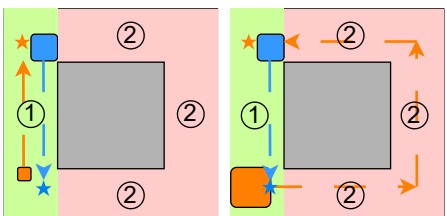

Figure 1: The partially observable bridge crossing task, where two agents (blue square and orange square) with changing physiques in different episodes (left and right figure) want to arrive at their destinations (stars with corresponding colors) through passageways 1 or 2. In this task, naively learning from a full observation expert policy would result in suboptimal partially observable policies due to the *asymmetric learning failure* (Warrington et al., 2021).

ing from such a centralized agent could lead to agents jamming on the same passage, as the partially observable agents cannot access the other agent's body size. In contrast, the optimal partially observable policies should ideally ensure that each agent consistently selects distinct passageways to avoid collision. This *asymmetric learning failure* is a prevalent issue in MARL due to the partial observability nature of MAS. While a few works have studied similar challenges in the context of single-agent RL (Walsman et al., 2023), it is worth noting that this issue within the MARL domain has not been thoroughly investigated to the best of our knowledge.

In this paper, we propose *correlated policy factorization*, dubbed AgentMixer, to tackle the above two challenges and achieve CE among agents in a fully decentralized way. Firstly, we propose a novel framework, named *Policy Modifier* (PM), to model the correlated joint policy, which takes as input decentralized partially observable policies and the state information and outputs the modified policies. Consequently, PM acts as an *observer* from the CE perspective and the modified policies form a correlated joint policy. Further, to mitigate the *asymmetric learning failure* when learning decentralized partially observable policies from the correlated joint fully observable policy, we then introduce a novel mechanism called *Individual-Global-Consistency* (IGC), which keeps consistent modes between individual policies and joint policy while allowing correlated exploration in joint policy. Theoretically, we prove that AgentMixer converges to $\epsilon$-approximate Correlated Equilibrium. Experimental results on various benchmarks confirm its strong empirical performance against current state-of-the-art MARL methods.

## 2 RELATED WORK

Modeling complex correlations among agents has been attracting a growing amount of attention in recent years. The centralized training decentralized execution (CTDE) paradigm has demonstrated its success in cooperative multi-agent domain (Lowe et al., 2017; Rashid et al., 2020a; Yu et al., 2022). Centralized training with additional global information makes agents cooperate better while decentralized execution enables distributed deployment.

**Value decomposition.** Value decomposition methods decompose the joint Q-function into individual utility functions following different interpretations of Individual-Global-Maximum (IGM) (Son et al., 2019), i.e., the consistency between optimal local actions and optimal joint action. VDN (Sunehag et al., 2017) and QMIX (Rashid et al., 2020b) decomposes the joint action-value function by additivity and monotonicity respectively. QTRAN (Son et al., 2019), WQMIX (Rashid et al.,

2020a) and QPLEX (Wang et al., 2021a) introduce additional components to enhance the expressive capability of value decomposition. To enhance coordination, MAVEN (Mahajan et al., 2019) introduces committed exploration among agents into QMIX, while DCG (Boehmer et al., 2020) models the interactions between agents with a coordination graph. Recent works delve into applying value decomposition to actor-critic methods (Su et al., 2021; Zhang et al., 2021). VDACs (Su et al., 2021), FACMAC (Peng et al., 2021) and DOP (Wang et al., 2021b) combine value decomposition to compute policy gradient with a centralized but factored critic. Zhang et al. (2021); Wang et al. (2023) derive joint soft-Q-function decomposition according to independent and conditional policy factorization respectively.

**Policy factorization.** Existing approaches commonly assume the independence of agents' policies, modeling the joint policy as the Cartesian Product of each agent's fully independent policy (Yu et al., 2022; Zhang et al., 2021; Kuba et al., 2021). However, such an assumption lacks in modeling complex correlations as it constrains the expressiveness of the joint policy and limits the agents' capability to coordinate. In contrast, some recent works (Wang et al., 2023; Wen et al., 2022; Fu et al., 2022) explicitly take the dependency among agents by presenting the joint policy in an auto-regressive form based on the chain rule (Box et al., 2015). MAT (Wen et al., 2022) casts MARL into a sequence modeling problem and introduces Transformer (Vaswani et al., 2017) to generate actions. Wang et al. (2023) extends FOP Zhang et al. (2021) with auto-regressive policy factorization. However, the lack of restrictions on dependent and independent policies may lead to inconsistencies. ACE (Li et al., 2023) transforms multi-agent Markov Decision Process (MMDP) (Littman, 1994) into a single-agent Markov Decision Process (MDP) (Feinberg & Shwartz, 2012), which implicitly models the auto-regressive joint policy. Despite the merits of the auto-regressive model, the fixed execution order and explicitly constrained representation limit the feasible joint policy space. Inspired by Correlated Equilibrium (Maschler et al., 2013), SIC (Chen et al., 2022) introduces a coordination signal to achieve richer classes of the joint policy and maximizes the mutual information (Kim et al., 2020) between the signal and the joint policy, which is close to MAVEN. Correlated Q-learning (Greenwald & Hall, 2003) generalizes Nash Q-learning (Hu & Wellman, 2003) based on CE and proposes several variants to resolve the equilibrium selection problem (Samuelson, 1997). Similarly, Schroeder de Witt et al. (2019) learns a hierarchical policy tree based on a shared random seed. Sheng et al. (2023) and Wen et al. (2019) learn coordinated behavior with recursive reasoning. However, most existing work focuses on fully observable settings or violates the decentralized execution requirement.

Moreover, existing approaches rarely study the issues arising from the use of asymmetric information (Warrington et al., 2021) in CTDE, that is, the joint fully observable critic or agent has access to information unavailable to the partially observable agents. In this paper, we study how to factorize the correlated joint fully observable policy into decentralized policies under partial observability.

## 3 PRELIMINARIES

### 3.1 DECENTRALIZED PARTIALLY OBSERVABLE MARKOV DECISION PROCESSES

In this work, we model a fully cooperative multi-agent game with $N$ agents as a *decentralized partially observable Markov decision process* (Dec-POMDP) (Oliehoek & Amato, 2016), which is formally defined as a tuple $\mathcal{G} = (\mathcal{N}, \mathcal{S}, \mathcal{O}, \mathbb{O}, \mathcal{A}, \mathcal{T}, \Omega, R, \gamma, \rho_0)$. $\mathcal{N} = \{1, \ldots, N\}$ is a set of agents, $s \in \mathcal{S}$ denotes the state of the environment and $\rho_0$ is the distribution of the initial state. $\mathcal{A} = \prod_{i=1}^{N} A^i$ is the joint action space, $\mathbb{O} = \prod_{i=1}^{N} O^i$ is the set of joint observations. At time step $t$, each agent $i$ receives an individual partial observation $o_t^i \in O^i$ given by the observation function $\mathcal{O} : (a_t, s_{t+1}) \mapsto P(o_{t+1}|a_t, s_{t+1})$ where $a_t, s_{t+1}$ and $o_{t+1}$ are the joint actions, states and joint observations respectively. Each agent $i$ uses a stochastic policy $\pi^i(a_t^i|h_t^i, \omega_t^i)$ conditioned on its action-observation history $h_t^i = (o_0^i, a_0^i, \ldots, o_{t-1}^i, a_{t-1}^i)$ and a random seed $\omega_t^i \in \Omega_t$ to choose an action $a_t^i \in A^i$. A belief state $b(s_t|h_t)$ is a sufficient statistic for joint history $h_t$, as an estimate of the underlying state $s_t$. Actions $a_t$ drawn from joint policy $\pi(a_t|s_t, \omega_t)$ conditioned on state $s_t$ and joint random seed $\omega_t = (\omega_t^1, \ldots, \omega_t^N)$ change the state according to transition function $\mathcal{T} : (s_t, a_t^1, \ldots, a_t^N) \mapsto P(s_{t+1}|s_t, a_t^1, \ldots, a_t^N)$. All agents share the same reward $r_t = R(s_t, a_t^1, \ldots, a_t^N)$ based on $s_t$ and $a_t$. $\gamma$ is the discount factor for future rewards. The goal of agents is to maximize the expected total reward, $\mathcal{J}(\pi) = \mathbb{E}_{s_0, a_0, \ldots} [\sum_{t=0}^{\infty} \gamma^t r_t]$, where $s_0 \sim \rho_0(s_0), a_t \sim \pi(a_t|s_t, \omega_t)$.

### 3.2 EQUILIBRIUM NOTIONS

We first define a *joint (potentially correlated) policy* as $\pi = \pi^1 \odot \pi^2 \cdots \odot \pi^N$. We also denote $\pi^{-i} = \pi^1 \odot \cdots \pi^{i-1} \odot \pi^{i+1} \odot \cdots \odot \pi^N$ to be the joint policy excluding the $i^{\text{th}}$ agent. A *product policy* is denoted as $\pi = \pi^1 \times \pi^2 \cdots \times \pi^N$ if the distribution of drawing each seed $\omega_t^i$ for different agents is independent. We define the value function $V_{\pi^i, \pi^{-i}}^i(s)$ as the expected returns under state $s$ that $i^{\text{th}}$ agent will receive if all agents follow joint policy $\pi = (\pi^i, \pi^{-i})$:

$$V_{\pi^i, \pi^{-i}}^i(s) = \mathbb{E}_{a_{0:\infty}^i \sim \pi^i, a_{0:\infty}^{-i} \sim \pi^{-i}, s_{1:\infty} \sim \mathcal{T}}[\Sigma_{t=0}^\infty \gamma^t r_t | s_0 = s]. \tag{1}$$

A *strategy modification* for the $i^{\text{th}}$ agent is a map $f^i : A^i \mapsto A^i$, which maps from the action set to itself. We can define the resulting policy by applying the map on $\pi^i$ as $f^i \diamond \pi^i$.

With the definition above, we can accordingly define the solution concepts.

**Definition 1** ($\epsilon$-approximate Nash Equilibrium). *A **product** policy $\pi_*$ is an $\epsilon$-approximate Nash Equilibrium (NE) if for for all $i \in \mathcal{N}$ and any $\epsilon \geq 0$:*

$$V_{\pi_*^i, \pi_*^{-i}}^i(s) \geq \max_{\pi^i} V_{\pi^i, \pi_*^{-i}}^i(s) - \epsilon. \tag{2}$$

**Definition 2** ($\epsilon$-approximate Coarse Correlated Equilibrium). *A **joint** policy $\pi_*$ is an $\epsilon$-approximate Coarse Correlated Equilibrium (CCE) if for for all $i \in \mathcal{N}$ and any $\epsilon \geq 0$:*

$$V_{\pi_*^i, \pi_*^{-i}}^i(s) \geq \max_{\pi^i} V_{\pi^i, \pi_*^{-i}}^i(s) - \epsilon. \tag{3}$$

The only difference between Definition 1 and Definition 2 is that an NE has to be a product policy while a CCE can be correlated.

**Definition 3** ($\epsilon$-approximate Correlated Equilibrium). *A joint policy $\pi_*$ is an $\epsilon$-approximate Correlated Equilibrium (CE) if for for all $i \in \mathcal{N}$ and any $\epsilon \geq 0$:*

$$V_{\pi_*^i, \pi_*^{-i}}^i(s) \geq \max_{f^i} V_{(f^i \diamond \pi_*^i) \odot \pi_*^{-i}}^i(s) - \epsilon. \tag{4}$$

It is also worth noting that an NE is always a CE, and a CE is always a CCE.

## 4 METHOD

In this work, we propose AgentMixer to achieve correlated policy factorization. The proposed method consists of two main components: *Policy Modifier* that models correlated joint fully observable policy and *Individual-Global-Consistency* that leverages the resulting joint policy for learning the individual policies while mitigating the *asymmetric information issue*.

### 4.1 POLICY MODIFIER

To efficiently introduce correlation among agents, we propose *Policy Modifier*, a novel framework based entirely on multi-layer perceptrons (MLPs) (see Appendix A), which contains two types of MLP layers (Tolstikhin et al., 2021): *agent-mixing MLPs* and *channel-mixing MLPs*. The agent-mixing MLPs allow inter-agent communication; they operate on each channel of the feature independently. The channel-mixing MLPs allow intra-agent information fusion; they operate on each agent independently. These two types of layers are interleaved to enable the interaction among agents and the correlated representation of the joint policy. Specifically, agent- and channel-mixing can be written as follows:

$$\begin{aligned} H_{\text{agent}} &= H_{\text{input}} + W_{\text{agent}}^{(2)} \sigma(W_{\text{agent}}^{(1)} \text{LayerNorm}(H_{\text{input}})), \\ H_{\text{channel}} &= H_{\text{agent}} + \sigma(W_{\text{channel}}^{(1)} \text{LayerNorm}(H_{\text{agent}})) W_{\text{channel}}^{(2)}, \end{aligned} \tag{5}$$

where $H_{\text{input}}$ is a concatenation of state features and individual policies features and W denotes fully connected layers. Then, the output of PM will be combined with individual policies to generate the correlated joint policy, denoted as $\text{PM}([\pi^i]_{i=1}^N) = ((f^1 \diamond \pi^1), \cdots, (f^N \diamond \pi^N)) = (f^1 \diamond \pi^1) \odot (f^2 \diamond \pi^2) \cdots \odot (f^N \diamond \pi^N)$, where $f$ denotes a *strategy modification*. Consequently, PM maps the individual policies into a correlated joint policy by introducing dependencies among agents.

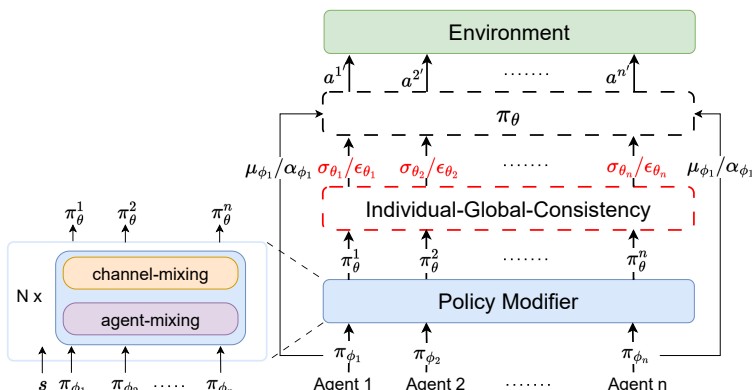

Figure 2: AgentMixer contains two components: 1) *Policy Modifier* takes the individual partially observable policies and state as inputs and produces correlated joint fully observable policy as outputs, and 2) *Individual-Global-Consistency* keeps the mode consistency among the joint policy and individual policies.

## 4.2 INDIVIDUAL GLOBAL CONSISTENCY

With the resulting correlated joint fully observable policy generated by PM, we can easily adopt different single-agent algorithms to get an (sub-)optimal correlated joint fully observable policy. To fulfill decentralized execution, we further ask a question:

**Question 1: Can we just derive the decentralized partially observable policies by distilling the learned (sub-)optimal correlated joint fully observable policy?**

In this section, we take several steps to provide a negative answer to the above research question. We begin by defining the joint policy and product policy as $\pi_\theta(a|s)$ and $\pi_\phi(a|b)$ respectively. Let the joint occupancy, $\rho^\pi(s, b)$, as the (improper) marginal state-belief distribution induced by a policy $\pi$: $\rho^\pi(s, b) = \sum_{t=0}^\infty \gamma^t P(s_t = s, b_t = b|\pi)$. Then, the marginal state distribution and marginal belief distribution induced by $\pi$ are denoted as $\rho^\pi(s) = \int_b \rho^\pi(s, b)db$ and $\rho^\pi(b) = \int_s \rho^\pi(s, b)ds$ respectively. To distill the joint policy $\pi_\theta(a|s)$ into the product policy $\pi_\phi(a|b)$, previous work (Ye et al., 2022) leverage imitation learning (Ross et al., 2011), i.e., optimizing the asymmetric distillation objective:

$$\mathbb{E}_{\rho^{\pi_\beta}(s,b)}\left[D_{\mathrm{KL}}\left(\pi_\theta(a|s) \| \pi_\phi(a|b)\right)\right], \text{where } \pi_\beta(s, b) = \beta\pi_\theta(a|s) + (1 - \beta)\pi_\phi(a|b). \quad (6)$$

$\pi_\beta$ is a mixture of the joint policy $\pi_\theta(a|s)$ and the product policy $\pi_\phi(a|b)$. The coefficient $\beta$ is annealed to zero during training. This avoids compounding error which grows with time horizon (Ross & Bagnell, 2010).

We then show that the optimal product policy defined by this objective can be expressed as posterior inference over state conditioned on the joint policy:

**Definition 4** (Implicit product policy). *For any correlated joint fully observable policy $\pi_\theta$ and any product partially observable behavioral policy $\pi_\psi$, we define $\hat{\pi}_\theta^\psi$ as the implicit product policy of $\pi_\theta$ under $\pi_\psi$ as:*

$$\hat{\pi}_\theta^\psi = \mathbb{E}_{\rho^{\pi_\psi}(s|b)}\left[\pi_\theta(a|s)\right], \quad (7)$$

*Such posterior inference has a fixed point (Warrington et al., 2021), i.e., $\pi_\psi = \hat{\pi}_\theta^\psi$, and we refer to this product policy as the implicit product policy of $\pi_\theta$, denoted as $\hat{\pi}_\theta$.*

Implicit product policy is defined as a posterior inference procedure, marginalizing the conditional occupancy $\rho^{\pi_\psi}(s|b)$. Since the observations/belief may not contain information to distinguish two different latent states, the $\rho^{\pi_\psi}(s|b)$ is a stochastic distribution, and the implicit product policy is the average of the fully observable policy. Suppose a scenario where the agent learns to cross the ice while avoiding the pits in the middle of the ice. The fully observable policy which can observe the location of the pits will choose safer routes that avoid the pits, i.e., both sides of the ice. However, according to 7, the implicit policy that is not informed of the pit locations will take an average path

of those safe routes, despite the danger of pits. The key insight is that directly imitating the fully observable policy will cause *asymmetric learning failure*.

We show that the solution to the asymmetric distillation objective in 6 is equivalent to the implicit product policy 7 in Appendix B. However, the implicit product policy requires marginalizing the conditional occupancy $\rho^\pi(s|b)$, which is intractable. Therefore, we can introduce a variational implicit product policy, $\pi_\eta$, as a proxy to the implicit product policy, which can be learned by minimizing the following objective:

$$\mathbb{E}_{\rho^{\pi_\psi}(s,b)}\left[D_{\mathrm{KL}}\left(\pi_\theta(a|s) \parallel \pi_\eta(a|b)\right)\right]. \tag{8}$$

Under sufficient expressiveness and exact updates assumptions, by setting $\pi_\psi = \pi_\eta$, updating 8 converges to the fixed point, i.e., the implicit product policy (see Appendix B).

We now reason about the *asymmetric learning failure*. In order to guarantee the optimal product partially observable policy, the divergence between the joint policy and product policy should be strictly zero, which we denote as *identifiability*:

**Definition 5** (Identifiable policy pair). *Given a correlated joint fully observable policy $\pi_\theta$ and a product partially observable policy $\pi_\phi$, we define $\{\pi_\theta, \pi_\phi\}$ as an identifiable policy pair if and only if $\mathbb{E}_{\rho^{\pi_\phi}(s,b)}\left[D_{\mathrm{KL}}\left(\pi_\theta(a|s) \parallel \pi_\phi(a|b)\right)\right] = 0$.*

Identifiable policy pairs require that the product partially observable policy can exactly recover the correlated joint fully observable policy. *Identifiability* then requires the optimal correlated joint fully observable policy and the corresponding implicit product policy to form an identifiable policy pair. Using *identifiability*, we can then prove that, given an optimal correlated joint fully observable policy, optimizing the asymmetric distillation objective is guaranteed to recover an optimal product partially observable policy:

**Theorem 1** (Convergence of asymmetric distillation). *Given an optimal correlated joint fully observable policy $\pi_{\theta^*}$ being identifiability, the iteration defined by:*

$$\eta_{k+1} = \arg\min_\eta \mathbb{E}_{\rho^{\pi_{\eta_k}}(s,b)}\left[D_{\mathrm{KL}}\left(\pi_{\theta^*}(a|s) \parallel \pi_\eta(a|b)\right)\right], \tag{9}$$

*converges to $\pi_{\eta^*}(a|b)$ that defines an optimal product partially observable policy, as $k \to \infty$.*

*Proof.* See Appendix B for detailed proof. $\square$

Theorem 1 shows that *identifiability* of the optimal joint policy defines a sufficient condition to guarantee the thorough distillation of the optimal joint fully observable policy into product partially observable policies. Unfortunately, the *identifiability* imposes a strong limitation on the applicability of asymmetric distillation. Hereby, we can conclude a negative answer to the **Question 1**. Therefore, instead of naively applying distillation on the learned joint policy, we simultaneously learn the correlated joint fully observable policy and its product partially observable counterpart. We will show that the interleaving of the two learning processes moves the product partially observable policy closer to Correlated Equilibrium, i.e., the optimal product partially observable policy.

We now use the insight from Theorem 1 and the definition of *identifiability* to define *Individual-Global-Consistency* (IGC), which keeps consistent modes between product partially observable policy and correlated joint fully observable policy.

**Definition 6** (IGC). *For a correlated joint fully observable policy $\pi_\theta(a|s)$, if there exist product partially observable policy $\pi_\phi(a|b) = \pi_{\phi^1}(a|h^1) \times \pi_{\phi^2}(a|h^2) \cdots \times \pi_{\phi^N}(a|h^N)$, such that the following holds:*

$$Mo(\pi_\theta) = \begin{pmatrix} Mo(\pi_{\phi^1}) \\ \vdots \\ Mo(\pi_{\phi^N}) \end{pmatrix}, \tag{10}$$

*where $Mo(\cdot)$ denotes the mode of distribution. Then, we say that $\pi_\phi(a|b)$ satisfy IGC.*

IGC enables the actions that occur most frequently in the joint policy and the product policy to be equivalent. Crucially, IGC minimizes the divergence between the two policies while allowing correlated exploration in the joint policy. Surprisingly, one may find that IGC and IGM are equivalent as monotonicity and mode consistency are similar.

### 4.2.1 IMPLEMENTATION OF IGC

In order to preserve IGC, we adopt the method of disentanglement between exploration and exploitation to decompose the joint policy into two components: one for the mode (exploitation) and the other for the deviation (exploration). Then, IGC can be enforced through an equality constraint on the relationship between the mode of joint policy and individual policies. Based on this disentanglement, agents are able to coordinate their exploration through the centralized policy. In practice, we divide the implementation of IGC into two categories: continuous action space and discrete action space.

**Continuous Case:** In this case, we assume the continuous action policy of agent $i$ as a Gaussian distribution with mean $\mu_{\phi^i}$ and standard deviation $\sigma_{\phi^i}$: $\pi_{\phi^i}(a|h^i) = \mathcal{N}(\mu_{\phi^i}(h^i), \sigma^2_{\phi^i}(h^i))$. Since the mode of a Gaussian distribution is equal to the mean, we set the mean of joint policy as the collection of individual policies while the standard deviation is generated by PM: $\pi_\theta(a|s) = \mathcal{N}(([\mu_{\phi^i}]^N_{i=1}), \sigma^2_\theta(s))$.

**Discrete Case:** In this case, we denote the discrete action policy of agent $i$ as a categorical distribution parameterized by probabilities $\alpha_{\phi^i}$:

$$\pi_{\phi^i}(a|h^i) = Cat(\alpha_{\phi^i}(h^i)) = \text{softmax}(\alpha_{\phi^i}(h^i)), \sum_{k=1}^{K} \alpha^k_{\phi^i}(h^i) = 1 \tag{11}$$

The mode of a categorical distribution is the most common category, the category with the highest frequency. However, it is tricky to promote cooperative exploration while preserving the mode consistency. Fortunately, Gumbel-Softmax distribution (Jang et al., 2017) provides another perspective, where we explicitly disentangle exploration and mode. Specifically, we define the joint policy as:

$$\pi_\theta = \begin{pmatrix} \text{softmax}((\epsilon^1_\theta + \log\alpha_{\phi^1})/\tau^1) \\ \vdots \\ \text{softmax}((\epsilon^N_\theta + \log\alpha_{\phi^N})/\tau^N) \end{pmatrix}, \tag{12}$$

where $\tau$ is a temperature hyperparameter and $\epsilon_\theta$ is sampled using inverse transform sampling by generating $u_\theta \in (0, 1)$ with sigmoid function and computing $\epsilon_\theta = -\log(-\log(u_\theta))$. Note that when the temperature approaches 0, the joint policy degrades to the collection of individual policies.

### 4.3 CONVERGENCE OF AGENTMIXER

Together with PM, we can view the learning of the correlated joint fully observable policy as a single-agent RL problem where abundant single-agent methods with theoretical guarantees of convergence and performance exist. Specifically, AgentMixer is trained end-to-end to maximize the following objective:

$$\mathcal{J}(\pi_\theta) = \mathbb{E}_{s\sim\rho^{\pi_\theta}, a\sim\pi_\theta}\left[\sum_{t=0}^{\infty} \gamma^t r_t\right], \text{subject to IGC, where} \pi_\theta = \text{PM}([\pi_{\phi^i}]^N_{i=1}). \tag{13}$$

**Theorem 2** (Convergence of AgentMixer). *The product partially observable policy generated by AgentMixer is a $\epsilon$-CE.*

*Proof.* For proof see Appendix B. □

With Theorem 3, we are ready to present the learning framework of AgentMixer, as illustrated in Figure 2, which consists of two main components: *Policy Modifier* and *Individual-Global-Consistency*. Specifically, PM acts as an observer who takes a holistic view and recommends that each agent follow her instructions. IGC then requires the agents to be obligated to follow the recommendations they receive. We provide the pseudo-code for AgentMixer in Appendix C. AgentMixer can benefit from a variety of strong single-agent algorithms, such as TD3 (Fujimoto et al., 2018), PPO (Schulman et al., 2017b), and SAC (Haarnoja et al., 2019). In this work, our implementation of AgentMixer follows PPO (Schulman et al., 2017b).

## 5 EXPERIMENTS

We compare our method with MAPPO (Yu et al., 2022), HAPPO (Kuba et al., 2021), MAT-Dec (Wen et al., 2022), MAVEN (Mahajan et al., 2019) and MACPF (Wang et al., 2023). An extensive evaluation is performed on both an illustrative matrix game (Lauer & Riedmiller, 2000) and two popular MARL benchmarks, Multi-Agent MuJoCo (Peng et al., 2021) (*MA-MuJoCo*) with continuous action space and *SMAC-v2* (Ellis et al., 2022) with discrete action space. More results and experimental details on these tasks are included in Appendix E.

### 5.1 CLIMBING MATRIX GAME

The climbing matrix game (Lauer & Riedmiller, 2000) has the payoff shown in the left of Figure 3. In this task, there are two agents to select the column and row index of the matrix respectively. The goal is to select the maximal element in the matrix. Although stateless and with simple action space, *Climbing* is difficult to solve via independent learning, as the agents need to coordinate among two optimal joint actions. The right of Figure 3 shows that the almost compared baselines converge to a local optimum while only AgentMixer and MAT successfully learn the optimal policy. This is reasonable, as in MAPPO, HAPPO, and MAT-Dec, agents are fully independent of each other when making decisions, they may fail to coordinate their actions, which eventually leads to a sub-optimal joint policy. While with an explicit external coordination signal, MAVEN only finds the optima by chance. For MAT, since it learns a centralized autoregressive policy, the second agent thus takes as input the first agent's action. It is not a surprise that MAT converges to the highest return due to using a centralized policy. In contrast, thanks to the introduced IGC mechanism, AgentMixer successfully learns fully decentralized optimal policies from the optimal correlated joint policy generated by the *Policy Modifier* (PM) module.

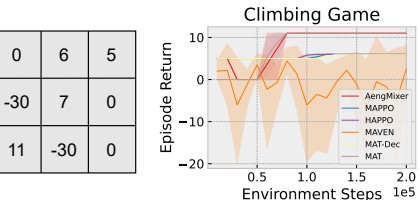

Figure 3: Left: the *Climbing* matrix game; right: the performance comparison.

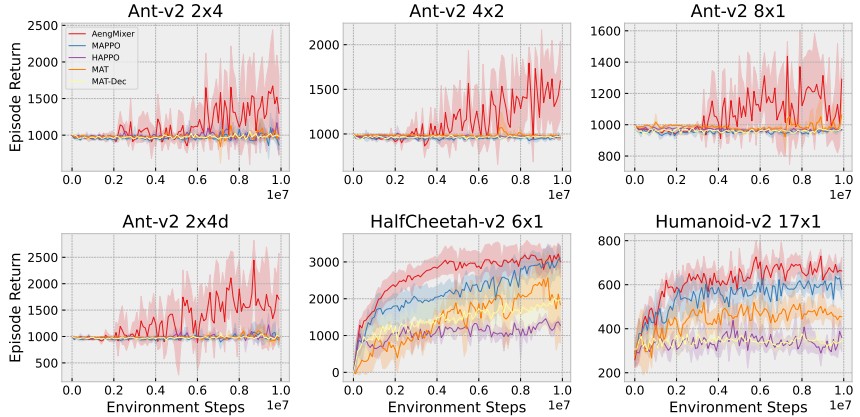

Figure 4: Performance comparison on multiple Multi-Agent MuJoCo tasks.

### 5.2 CONTINUOUS ACTIONS SPACES: *MA-MuJoCo*

As in the full observation setting, previous methods have shown near-optimal performance in the *MA-MuJoCo* tasks (Kuba et al., 2021; Wen et al., 2022), we instead set $obsk = 0$ for all the tasks, which means that each agent can only observe its own joint information and satisfies better the partial observability nature in MARL. We show the performance comparison against the baselines in Figure 4. We can see that AgentMixer enjoys superior performance over those baselines. The superiority of our method is highlighted especially in Ant-v2 tasks, where partial observability poses a critical challenge as the local observations of each agent (leg) of the ant are quite similar and make it hard to estimate the necessary state information for coordination. In these tasks, while other algorithms, even the centralized MAT, fail to learn any meaningful joint policies, AgentMixer outperforms the

baselines by a large margin. These results show that AgentMixer can effectively exploit asymmetric information to mitigate the challenges incurred by severe partial observability.

### 5.3 DISCRETE ACTION SPACES: *SMAC-v2*

Compared to the StarCraft Multi-Agent Challenge (*SMAC*), we instead evaluate our method on the more challenging *SMAC-v2* benchmark which is designed with higher randomness. As shown in Figure 5, we generally observe that AgentMixer achieves comparable performance compared with the baselines. Note that even centralized MAT performs similarly to other decentralized counterparts.

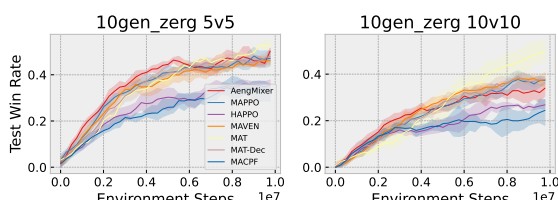

Figure 5: Comparison of the mean test win rate on SMACv2.

### 5.4 ABLATION RESULTS

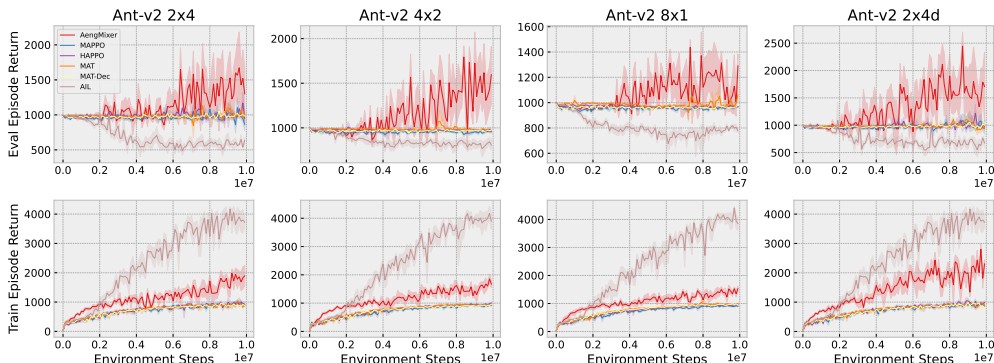

Figure 6: Ablations on Ant-v2. The large performance gap can be seen between training and testing on AIL, which is caused by *addressing asymmetric learning failure*. Other baselines fail to learn any effective policies, while AgentMixer obtains superior performance.

To examine the effectiveness of AgentMixer in *addressing asymmetric learning failure*, we perform ablation experiments by adding an imitation learning baseline, asymmetric imitation learning (AIL) (Warrington et al., 2021), which uses PPO, conditioned on full state information, to supervise learning decentralized policies, conditioned on partial information. As shown in Figure 6, due to *asymmetric learning failure*, AIL performs poorly in evaluation, although it achieves superior performance in training. In contrast, AgentMixer couples the learning of the centralized policy and decentralized policies such that partially observed policies can perform consistently with the fully observed policy.

## 6 CONCLUSION

In order to achieve coordination among partially observable agents, this paper presents a novel framework named AgentMixer which enables *correlated policy factorization* and provably converges to $\epsilon$-approximate Correlated Equilibrium. AgentMixer consists of two key components: 1) the *Policy Modifier* that takes all the initial decisions from individual agents and composes them into a correlated joint policy based on the full state information; 2) the *Individual-Global-Consistency* which mitigates the *asymmetric learning failure* by preserving the consistency between individual and joint policy. Surprisingly, IGC and IGM can be considered as parallel works of policy gradient-based and value-based methods respectively. We will study the transformation between IGC and IGM in future work. We extensively evaluate the proposed method on both an illustrative matrix game and two popular MARL benchmarks. The experiments demonstrate that our method outperforms strong baselines in most tasks and achieves comparable performance in the rest.

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

## A    DETAIL STRUCTURE OF POLICY MODIFIER

Figure 7 depicts the macro-structure of *Policy Modifier*. It accepts the state and policies of agents as input. Specifically, *Policy Modifier* two MLP blocks. The first one is the *agent-mixing MLPs*: it acts on columns of input. The second one is the *channel-mixing MLP*: it acts on rows of the output of *agent-mixing MLPs*.

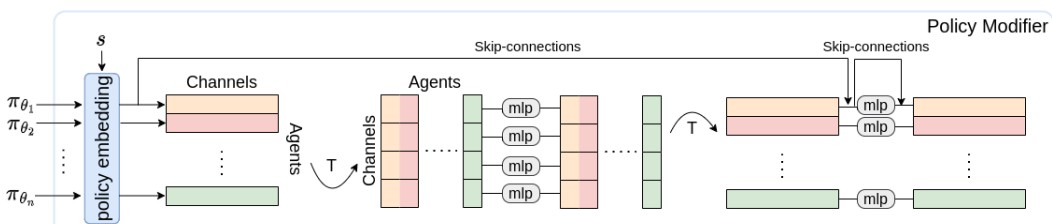

Figure 7: Policy Modifier consists of policy embedding layer, agent-mixing MLP and channel-mixing MLP.

## B    ADDITIONAL PROOFS

### B.1    PROOF OF LEMMA 1

**Lemma 1** (Asymmetric distillation solution). *theoremad For any correlated joint fully observable policy $\pi_\theta$ and fixed product partially observable behavioral policy $\pi_\psi$, the implicit product policy $\hat{\pi}_\theta^\psi$, defined in Definition 4, minimizes the asymmetric distillation objective:*

$$\hat{\pi}_\theta^\psi = \arg\min_\pi \mathbb{E}_{\rho^{\pi_\psi}(s,b)} \left[ D_{\mathrm{KL}} \left( \pi_\theta(a|s) \,\|\, \pi(a|b) \right) \right]. \tag{14}$$

*Proof.* Expanding the right-hand side:

$$\arg \min_{\pi} \mathbb{E}_{\rho^{\pi_\psi}(s,b)} \left[ D_{\mathrm{KL}} \left( \pi_\theta(a|s) \parallel \pi(a|b) \right) \right]$$

$$= \arg \min_{\pi} \mathbb{E}_{\rho^{\pi_\psi}(b)} \left[ \int_s \int_a \pi_\theta(a|s) \log(\frac{\pi_\theta(a|s)}{\pi(a|b)}) da \rho^{\pi_\psi}(s|b) ds \right],$$

$$= \arg \min_{\pi} \mathbb{E}_{\rho^{\pi_\psi}(b)} \left[ \int_s \mathcal{H}(\pi_\theta(\cdot|s)) \rho^{\pi_\psi}(s|b) ds \right] - \mathbb{E}_{\rho^{\pi_\psi}(b)} \left[ \int_s \int_a \pi_\theta(a|s) \log(\pi(a|b)) da \rho^{\pi_\psi}(s|b) ds \right],$$

where $\mathcal{H}(\cdot)$ is the entropy function,

$$= \arg \min_{\pi} \mathrm{const} - \mathbb{E}_{\rho^{\pi_\psi}(b)} \left[ \int_a \hat{\pi}_\theta^\psi(a|b) \log(\pi(a|b)) da \right],$$

note that we are free to set the const, so long as it remains independent of $\pi$,

$$= \arg \min_{\pi} \mathbb{E}_{\rho^{\pi_\psi}(b)} \left[ \int_a \hat{\pi}_\theta^\psi(a|b) \log(\hat{\pi}_\theta^\psi(a|b)) da - \int_a \hat{\pi}_\theta^\psi(a|b) \log(\pi(a|b)) da \right],$$

$$= \arg \min_{\pi} \mathbb{E}_{\rho^{\pi_\psi}(b)} \left[ D_{\mathrm{KL}} \left( \hat{\pi}_\theta^\psi(a|b) \parallel \pi(a|b) \right) \right].$$

$$(15)$$

Hence we conclude the proof. □

## B.2 PROOF OF CONVERGENCE OF ITERATIVE VARIATIONAL APPROXIMATION

We first introduce an assumption which simply states that the variational family is sufficiently expressive such that the implicit product policy can be recovered, and the implicit product policy is sufficiently expressive such that the optimal product partially observable policy can actually be found.

**Assumption 1** (Sufficiency of Variational Representations). *We assume that for any product behavioral policy, $\pi_\psi$, the variational family is sufficiently expressive such that any implicit product policy, $\hat{\pi}_\theta$, can be exactly recovered under the occupancy induced by the product behavioral policy. We also assume that there is an implicit product policy, $\hat{\pi}_\theta$, such that an optimal product partially observable policy can be represented, and thus there is a variational implicit product policy that can represent the optimal product partially observable policy under $\rho^{\pi_\psi}(b)$.*

We then introduce the lemma which shows that the solution to an iterative procedure actually converges to the solution of a single equivalent "static" optimization problem. This lemma allows us to solve the challenging optimization using a simple iterative procedure.

**Lemma 2** (Convergence of Iterative Variational Approximation). *Given the implicit product policy $\hat{\pi}_\theta$ and the corresponding variational approximation to $\hat{\pi}_\theta$, $\pi_\eta$, then under Assumption 1, the iterative procedure:*

$$\eta_{k+1} = \arg \min_{\eta} \mathbb{E}_{\rho^{\pi_{\eta_k}}(b)} \left[ D_{\mathrm{KL}} \left( \hat{\pi}_\theta(a|b) \parallel \pi_\eta(a|b) \right) \right], \textit{with } k \to \infty, \tag{16}$$

*converges to the solution to the optimization problem:*

$$\eta^* = \arg \min_{\eta} \mathbb{E}_{\rho^{\pi_\eta}(b)} \left[ D_{\mathrm{KL}} \left( \hat{\pi}_\theta(a|b) \parallel \pi_\eta(a|b) \right) \right]. \tag{17}$$

*Proof.* We begin by expressing the total variation between $\rho^{\pi_{\eta^*}}(b)$ and $\rho^{\pi_{\eta^k}}(b)$ at the $k^{\mathrm{th}}$ iteration:

$$D_{\mathrm{TV}}(\rho^{\pi_{\eta^*}}(b) \parallel \rho^{\pi_{\eta^k}}(b)) = \sup_b \left| \sum_{t=0}^{\infty} \gamma^t P(b_t = b | \pi_{\eta^*}) - \sum_{t=0}^{\infty} \gamma^t P(b_t = b | \pi_{\eta^k}) \right|,$$

$$= \sup_b | \sum_{t=0}^{k} \gamma^t P(b_t = b | \pi_{\eta^*}) + \sum_{t=k+1}^{\infty} \gamma^t P(b_t = b | \pi_{\eta^*}) \tag{18}$$

$$- \sum_{t=0}^{k} \gamma^t P(b_t = b | \pi_{\eta^k}) - \sum_{t=k+1}^{\infty} \gamma^t P(b_t = b | \pi_{\eta^k}) |.$$

We can then note that at the $k^{\mathrm{th}}$ iteration, the marginal belief distributions induced by $\pi_{\eta^*}$ and $\pi_{\eta^k}$ over the first $k$ iteration must be identical as the underlying dynamics are the same at the initial state

and belief state and we have exactly minimized the $D_{\mathrm{KL}}(\pi_{\eta^*} \parallel \pi_\eta)$. With the assumption that the maximum variation between the densities is bounded by $C$, we have:

$$
\begin{aligned}
&\sup_b |\sum_{t=0}^k \gamma^t P(b_t = b|\pi_{\eta^*}) + \sum_{t=k+1}^\infty \gamma^t P(b_t = b|\pi_{\eta^*}) \\
&\quad - \sum_{t=0}^k \gamma^t P(b_t = b|\pi_{\eta^k}) - \sum_{t=k+1}^\infty \gamma^t P(b_t = b|\pi_{\eta^k})|, \\
&= \sup_b |\sum_{t=k+1}^\infty \gamma^t (P(b_t = b|\pi_{\eta^*}) - P(b_t = b|\pi_{\eta^k}))|, \\
&\leq \sup_b |\sum_{t=k+1}^\infty \gamma^t C|, \\
&= C(\frac{1}{1-\gamma} - \frac{1-\gamma^{k+1}}{1-\gamma}), \\
&= C\frac{\gamma^{k+1}}{1-\gamma} = O(\gamma^k).
\end{aligned}
\tag{19}
$$

Hence, as $\gamma \in [0,1)$, the total variation between $\pi_{\eta^*}$ and $\pi_{\eta^k}$ converges to zero as $k \to \infty$. With this result and the expressiveness assumption, we complete the proof. $\qquad\square$

Similar to the proof of Lemma 2, we can derive the following result:

$$
\arg\min_\eta \mathbb{E}_{\rho^{\pi_\psi}(b)} \left[ D_{\mathrm{KL}} \left( \hat\pi_\theta(a|b) \parallel \pi_\eta(a|b) \right) \right] = \arg\min_\eta \mathbb{E}_{\rho^{\hat\pi_\theta}(b)} \left[ D_{\mathrm{KL}} \left( \hat\pi_\theta(a|b) \parallel \pi_\eta(a|b) \right) \right]. \tag{20}
$$

This result allows us to exchange the distribution under which we take expectations.

### B.3   PROOF OF THEOREM 1

With Assumption 1, Lemma 2, and the identifiability condition, we are ready to verify the convergence of asymmetric distillation.

**Theorem 1** (Convergence of asymmetric distillation). *Given an optimal correlated joint fully observable policy $\pi_{\theta^*}$ being identifiability, the iteration defined by:*

$$
\eta_{k+1} = \arg\min_\eta \mathbb{E}_{\rho^{\pi_{\eta_k}}(s,b)} \left[ D_{\mathrm{KL}} \left( \pi_{\theta^*}(a|s) \parallel \pi_\eta(a|b) \right) \right], \tag{9}
$$

*converges to $\pi_{\eta^*}(a|b)$ that defines an optimal product partially observable policy, as $k \to \infty$.*

*Proof.* We begin by considering the limiting behavior as $k \to \infty$:

$$
\begin{aligned}
\eta^* &= \lim_{k\to\infty} \arg\min_\eta \mathbb{E}_{\rho^{\pi_{\eta_k}}(s,b)} \left[ D_{\mathrm{KL}} \left( \pi_{\theta^*}(a|s) \parallel \pi_\eta(a|b) \right) \right], \\
&= \lim_{k\to\infty} \arg\min_\eta \mathbb{E}_{\rho^{\pi_{\eta_k}}(b)} \left[ D_{\mathrm{KL}} \left( \hat\pi_{\theta^*}(a|b) \parallel \pi_\eta(a|b) \right) \right], \text{(Lemma 1)} \\
&= \arg\min_\eta \mathbb{E}_{\rho^{\pi_\eta}(b)} \left[ D_{\mathrm{KL}} \left( \hat\pi_{\theta^*}(a|b) \parallel \pi_\eta(a|b) \right) \right], \text{(Lemma 2)} \\
&= \arg\min_\eta \mathbb{E}_{\rho^{\hat\pi_{\theta^*}}(b)} \left[ D_{\mathrm{KL}} \left( \hat\pi_{\theta^*}(a|b) \parallel \pi_\eta(a|b) \right) \right], \text{(exchange the distribution)} \\
&= \arg\min_\eta \mathbb{E}_{\rho^{\pi_{\phi^*}}(b)} \left[ D_{\mathrm{KL}} \left( \pi_{\phi^*}(a|b) \parallel \pi_\eta(a|b) \right) \right]. \text{(identifiability)}
\end{aligned}
\tag{21}
$$

Finally, under Assumption 1, the expected KL divergence can be exactly zero, which completes the proof. $\qquad\square$

### B.4   PROOF OF THEOREM 2

**Theorem 2** (Convergence of AgentMixer). *The product partially observable policy generated by AgentMixer is a $\epsilon$-CE.*

*Proof.* Since PM modifies the decentralized policies and generates the correlated joint policy, i.e., $\pi_\theta = ((f_\theta^1 \diamond \pi_{\phi^1}), \cdots, (f_\theta^N \diamond \pi_{\phi^N}))$, the RL procedure mentioned in 13 can be regarded as a single-agent RL problem. By leveraging Theorem 1 in TRPO (Schulman et al., 2015), we can

conclude that a sequence $(\pi_{\theta_k})_{k=1}^{\infty}$ of joint policies updated by 13 has the monotonic improvement property, i.e., $\mathcal{J}(\pi_{\theta_{k+1}}) \geq \mathcal{J}(\pi_{\theta_k})$. According to Bolzano-Weierstrass Theorem, the sequence of policies $(\pi_{\theta_k})_{k=1}^{\infty}$ exists at least one sub-optimal point $\pi_{\theta*}$. Let $\pi_{\bar{\theta}}$ be the optimal joint policy and $\epsilon = V_{\pi_{\bar{\theta}}}(s) - V_{\pi_{\theta*}}(s) \geq 0$. Given the value function defined in 1, we have:

$$V_{\pi_{\theta*}}(s) + \epsilon \geq V_{\pi_{\theta}}(s), \forall \pi_{\theta}. \tag{22}$$

Since optimizing $\pi_{\theta}$ is actually optimizing $f_{\theta}$, then we can obtain:

$$V_{((f_{\theta*}^1 \diamond \pi_{\phi^1}), \cdots, (f_{\theta*}^N \diamond \pi_{\phi^N}))}(s) + \epsilon \geq \max_{f_{\theta}} V_{((f_{\theta}^1 \diamond \pi_{\phi^1}), \cdots, (f_{\theta}^N \diamond \pi_{\phi^N}))}(s). \tag{23}$$

Applying IGC, which keeps the mode consistency between joint policy and product policy, yields:

$$((f_{\theta*}^1 \diamond \pi_{\phi^1}), \cdots, (f_{\theta*}^N \diamond \pi_{\phi^N})) = (\pi_{\phi^1}, \cdots, \pi_{\phi^N}). \tag{24}$$

Finally, by plugging 24 into 23:

$$V_{(\pi_{\phi^1}, \cdots, \pi_{\phi^N})}(s) \geq \max_{f_{\theta}} V_{((f_{\theta}^1 \diamond \pi_{\phi^1}), \cdots, (f_{\theta}^N \diamond \pi_{\phi^N}))}(s) - \epsilon. \tag{25}$$

which is exactly the $\epsilon$-CE defined in Definition 3. $\square$

## C  PSEUDO-CODE FOR AGENTMIXER

The pseudo-code of our method is shown in Algorithm C.

---
**Algorithm 1** AgentMixer

---
INITIALIZE Decentralized partially observable policies $\{\pi_{\phi^1}, \ldots, \pi_{\phi^N}\}$, a single agent algorithm A.
//Construct the joint policy:
$\pi_{\theta} = \mathrm{PM}([\pi_{\phi^i}]_{i=1}^N)$, subjected to IGC.
Run A on $\pi_{\theta}$.
RETURN $\{\pi_{\phi^1}, \ldots, \pi_{\phi^N}\}$.

---

## D  BASELINES AND MORE EXPERIMENTS

We compare our method with the baselines below including both algorithms with state-of-the-art performance and methods designed specifically to tackle the coordination problems.

**MAPPO** (Yu et al., 2022) applies PPO (Schulman et al., 2017a) to multi-agent settings and utilizes CTDE to learn critics based on the global state in order to stabilize the policy gradient estimation. Although with simple techniques, MAPPO has achieved tremendous empirical success in various multi-agent domains and can be a strong baseline for our method.

**HAPPO** (Kuba et al., 2021) performs sequential policy updates by utilizing other agents' newest policy under the CTDE framework and provably obtains the monotonic policy improvement guarantee as in single-agent PPO.

**MAT-Dec** (Wen et al., 2022) is the decentralized version of *MAT* which models the multi-agent decision process as a sequence-to-sequence generation problem with powerful transformer architecture (Vaswani et al., 2017). *MAT-Dec* relaxes the restriction of using other agents' actions but remains taking the full observations from other agents. Therefore, we remind that *MAT-Dec* uses full state information in experiments while our method and other baselines are limited by partial observation.

**MAVEN** (Mahajan et al., 2019) is proposed to improve the exploration of QMIX by introducing a latent space for hierarchical control. Compared to QMIX, MAVEN takes further advantage of CTDE through a *committed exploration* strategy.

**MACPF** (Wang et al., 2023) extends SAC (Haarnoja et al., 2019) into multi-agent settings and explicitly introduces auto-regressive dependency among agents.

**AIL** (Wang et al., 2023) naively distills partially observable agents' policies from the fully observable centralized policy. Although showing significant training performance, it suffers from the *asymmetric learning failure* problem and fails to perform well during execution with partial observation.

We summarize the different CTDE settings in Table 1. Note that although MAT and MAT-Dec sometimes show better performance than other methods, they take the full state information even during execution.

| Algorithm | MAPPO | HAPPO | MAT | MAT-Dec | MAVEN | MACPF | AIL | Ours |
|---|---|---|---|---|---|---|---|---|
| P. Ob. (execution) | ✓ | ✓ | ✗ | ✗ | ✓ | ✓ | ✓ | ✓ |
| C. Value (training) | ✓ | ✓ | ✓ | ✓ | ✓ | ✓ | ✓ | ✓ |
| C. Policy (training) | ✗ | ✗ | ✗ | ✗ | ✗ | ✗ | ✓ | ✓ |

Table 1: This table compares different settings of CTDE used in baselines with our methods, where *P. Ob.* denotes *partial observation*, *C. Value* means *centralized value* and *C. Policy* represents *centralized policy*. Note that only methods with *partial observation* during execution are fair for comparison. Although all the methods take advantage of CTDE by using a *centralized value* during training, only our method and AIL further employ *centralized policy* for training. Our method tackles the *asymmetric learning failure* problem in AIL and hence shows better performance.

### D.1 MORE EXPERIMENTS ON *MA-MuJoCo*

Inspired by mujoco tasks in the single-agent RL realm, *MA-MuJoCo* splits the joints of robots into different agents to enable decentralized control for MARL research. *MA-MuJoCo* allows different observation settings by changing the parameter of *obsk* which controls the number of neighbor joints each agent can observe.

We show the experiment results on more *MA-MuJoCo* tasks in Figure 8. Extended ablation study results are shown in Figure 9.

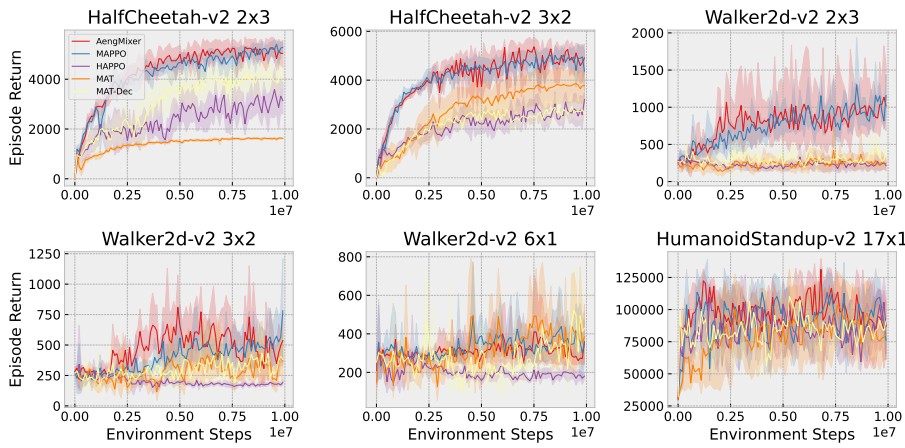

Figure 8: Performance comparison on multiple Multi-Agent MuJoCo tasks.

### D.2 MORE EXPERIMENTS ON SMAC-V2

The original StarCraft Multi-Agent Challenge (*SMAC*) has been shown to be not difficult enough, as an open-loop policy conditioned only on the timestep can achieve non-trivial win rates for many scenarios. To address these shortcomings, a new benchmark, SMACv2 was proposed to address SMAC's lack of stochasticity.

We compare a baseline, MAPPO_FULL, conditioned on full state information during evaluation. The results in Fig. 12 show that partially observable policies achieve similar performance as fully

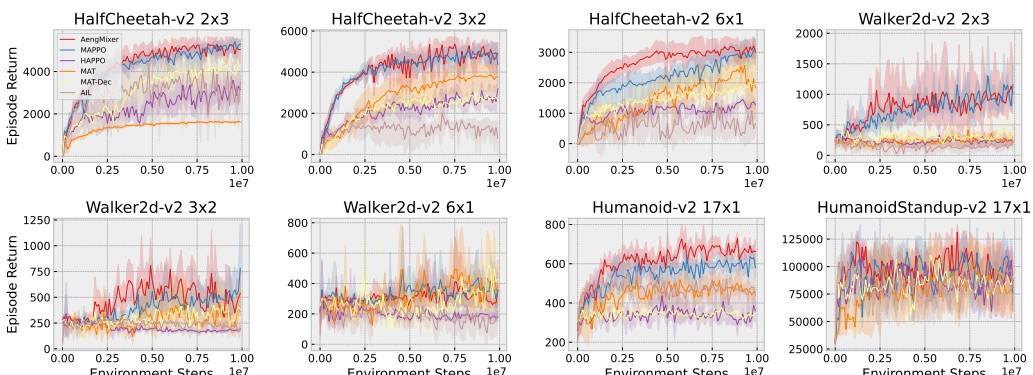

Figure 9: Ablations on multiple Multi-Agent MuJoCo tasks.

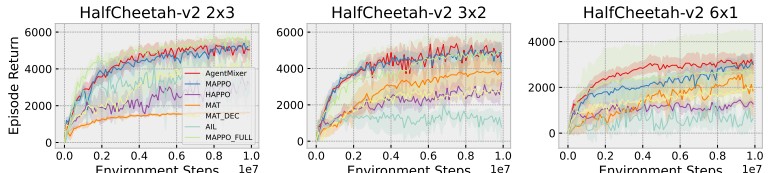

Figure 10: Ablations on multiple Multi-Agent MuJoCo tasks.

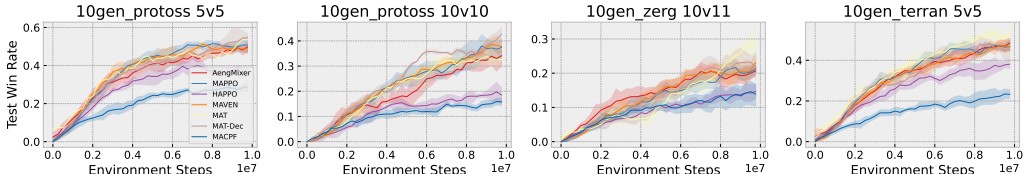

Figure 11: Comparison of the mean test win rate on SMACv2.

observable policies. This demonstrates that global information is not important for learning in the SMACv2 domain.

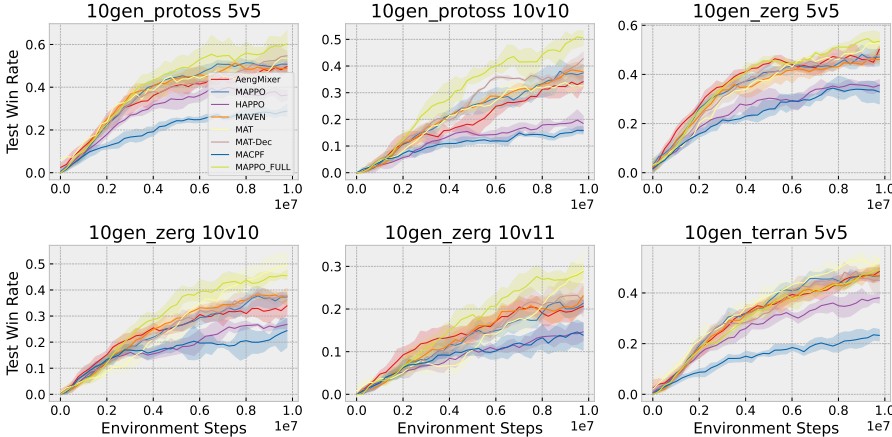

Figure 12: Ablations demonstrating the effect of full state information on SMACv2.

### D.3 MORE EXPERIMENTS ON PREDATOR-PREY

We use an environment similar to that described by Li et al. (2020) where agents are controlled to capture prey. If a prey is captured, the agents receive a reward of 10. However, the environment penalizes any single-agent attempt to capture prey with a penalty. Figure 13 shows the average return for test episodes for varying penalties.

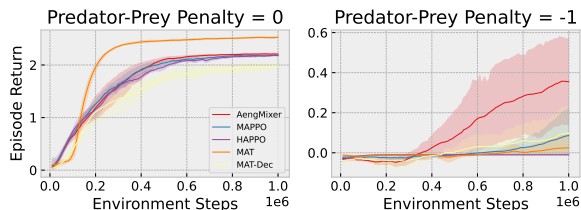

Figure 13: Performance comparison on Predator-Prey with different penalties for single-agent capture attempt.

## E   HYPER-PARAMETERS

For a fair comparison, the implementation of AgentMixer and the baselines are based on the implementation of MAPPO. We keep all hyper-parameters unchanged at the origin best-performing status. The proposed method and compared baselines are implemented into parameter independent version except MAT and MAT-Dec. The common and different hyper-parameters used for the baselines and AgentMixer across all domains are listed in Table 2-8 respectively.

### E.1   COMMON HYPER-PARAMETERS

We list the common hyper-parameters across all the domains in Table 2-5.

### E.2   MATRIX GAMES

We list the hyper-parameters used in matrix games in Table 6.

| Parameter | Value |
|---|---|
| agent-mixing hidden dim | 32 |
| channel-mixing hidden dim | 256 |
| mixer lr | 5e-5 |

Table 2: Unique hyper-parameters of AgentMixer.

| Parameter | Value |
|---|---|
| block number | 1 |
| head number | 1 |

Table 3: Unique hyper-parameters of MAT / MAT-Dec.

| Parameter | Value |
|---|---|
| noise dim | 2 |
| epsilon start | 1.0 |
| epsilon end | 1.0 |
| target update interval | 200 |

Table 4: Unique hyper-parameters of MAVEN.

| Parameter | Value |
|---|---|
| Training | |
| optimizer | Adam |
| optimizer epsilon | 1e-5 |
| weight decay | 0 |
| max grad norm | 10 |
| data chunk length | 1 |
| Model | |
| activation | ReLU |
| PPO | |
| ppo-clip | 0.2 |
| gamma | 0.99 |
| gae lambda | 0.95 |

Table 5: Common hyper-parameters used across all domains.

## E.3  SMACv2

We list the hyper-parameters used for each map of SMACv2 in Table 7.

## E.4  MA-MuJoCo

The hyper-parameters used for each task of MA-MuJoCo are listed in Table 8.

| Parameter | Value |
|---|---|
| Training | |
| actor lr | 5e-4 |
| critic lr | 5e-4 |
| entropy coef | 0.01 |
| Model | |
| hidden layer | 1 |
| hidden layer dim | 64 |
| PPO | |
| ppo epoch | 15 |
| ppo-clip | 0.2 |
| num mini-batch | 1 |
| Sample | |
| environment steps | 200000 |
| rollout threads | 50 |
| episode length | 200 |

Table 6: Common hyper-parameters used in matrix games.

| Parameter | Value |
|---|---|
| Training | |
| actor lr | 5e-4 |
| critic lr | 5e-4 |
| entropy coef | 0.01 |
| Model | |
| hidden layer | 1 |
| hidden layer dim | 64 |
| PPO | |
| ppo epoch | 5 |
| ppo-clip | 0.2 |
| num mini-batch | 1 |
| Sample | |
| environment steps | 10000000 |
| rollout threads | 50 |
| episode length | 200 |

Table 7: Common hyper-parameters used in the SMACv2.

| Parameter | Value |
|---|---|
| Training | |
| actor lr | 3e-4 |
| critic lr | 3e-4 |
| entropy coef | 0 |
| Model | |
| hidden layer | 2 |
| hidden layer dim | 64 |
| PPO | |
| ppo epoch | 5 |
| ppo-clip | 0.2 |
| num mini-batch | 1 |
| Sample | |
| environment steps | 10000000 |
| rollout threads | 40 |
| episode length | 100 |
| Environment | |
| agent obsk | 0 |

Table 8: Common hyper-parameters used in the MA-MuJoCo.

