# OpenReview forum: "AgentMixer: Multi-Agent Correlated Policy Factorization"
_ICLR.cc/2024/Conference — ICLR 2024 Conference Withdrawn Submission_

### Official Review · Reviewer_tDVs · 2023-10-23

**Soundness:** 2 fair
**Presentation:** 2 fair
**Contribution:** 2 fair
**Rating:** 5
**Confidence:** 4

**Summary:**

The goal of this paper is to learn agent policies that satisfy a correlated equilibrium, which can be executed in a fully decentralized manner, under partial observability for each agent. The authors propose the AgentMixer method. The main two aspect of the method are (1) the Policy Modifier, which takes the fully decentralized policies and state information, producing a correlated joint policy from the decentralized policies, and (2) a method to extract decentralized policies from the centralized policy, while ensuring the Individual Global Consistency (IGC) condition. The AgentMixer method follows the CTDE framework, assuming that a centralized controller can both observe the full state during training, and send joint actions to the environment.

**Strengths:**

I enjoyed reading this paper, which focuses on how we might represent an optimal joint policy in the form of a product policy (where the latter can be executed using decentralized execution. The motivation and proposed method, based on posterior inference (?) is also interesting. The paper provided a good view of the literature + state-of-the-art for value factorization work in MARL, and the selected experimental settings were challenging.

**Weaknesses:**

While the idea is interesting, the paper is weak in many aspects. The points below are listed in order of most to least important for the authors to fix.

- Incompletely specified method:
	1) Preserving the IGC is clearly an important part of the method. While the paper clearly defines IGC (Def 6) and discusses why it is important to preserve that condition, it doesn’t explain how the proposed method preserves IGC. The only provided explanation is Equation 13, which presents a centralized reward-maximization objective for MARL, subject to  the constraint that IGC is preserved. This is clearly not a sufficient explanation, and the objective isn't a novel insight. Arguably many value-factorization methods maximize this objective (e.g. VDN, QMIX, QTRAN, etc.) -- the crux is how IGC is maintained.

- Incomplete results + limited ablation studies:
	1) Figure 3, the curve for AengMixer is cut off. It is important to show the full curve, to let the reader verify the asymptotic behavior of the method.
	2) Overall, it seems there should be other evaluations. For example, it seems natural to have an ablation to study the degree of partial observability that the IGC can handle. Also, the authors should attempt using another method to generate the fully observable joint policy. What about directly running a single-agent RL algorithm on the joint action space (rather than the proposed AgentMixer architecture)?
- Weak improvement overall on both MAMujoco, and StarCraft
- Some missing references to prior work. For example, Greenwald et al.'s classic paper on [Correlated Q-learning](https://dl.acm.org/doi/10.5555/3041838.3041869)  should be added and discussed. The authors should also add a discussion of related work on handling partial observability in multi-agent systems.

- Notation  and clarity should be improved:
	1) It's confusing that there's no difference in notation or wording between Definition 1 and 2. Further, it seems worth defining correlated equilibria, and commenting on the distinction from a coarse correlated equilibria in more detail.
	2) The explanation of the policy mixer in Section 3.1 is very confusing. For example, what is W in equation 6? What is "f^I" in the paragraph after eq 6? How are the individual policies combined to form a joint policy?
	3) The key figure, Figure 2 should be much earlier in the paper.
	4) Typos: see "AgnetMixer" under Figure 2; "AengMixer" in Figures 3-5
	5) The placement of the related works section is very odd; usually it is the last section in the paper before the conclusion, or the second section in the paper (after introduction)
	6) Definition 4 involves a whole posterior inference procedure which is not described enough. Is this procedure purely for illustrative purposes? Or is it actually part of the method proposed by the paper? If the latter, the authors need to add a lot more explanation. The authors should add at least a paragraph of explanation for this, preferably also including an Algorithm detailing this procedure and a figure in the Appendix. For example, can you elaborate further on what pi_psi is and where does it come from? Also, the text states that the procedure given by Eq. (8) converges to a fixed point. Can the authors add references or proof?

**Questions:**

Please see the Weaknesses section for the most important questions. Some additional questions are below:
- Why does AgentMixer show the most significant performance improvements in Ant and not the other tasks?
- Does AgentMixer consistently solve relative overgeneralization problems that multi-agent policy gradient methods frequently suffers from? Perhaps this evaluation on RO domains might highlight the benefits of this method. The authors could try running experiments on the predator-prey w/punishment domain specified in [this paper](https://www.ifaamas.org/Proceedings/aamas2021/pdfs/p764.pdf).

---

> ### Author Response · Authors · 2023-11-17
> **Response to Reviewer tDVs**
>
> ## Response to Reviewer tDVs:
>
> We thank you for your kind comments and apologize for the unclear statements in our paper. We believe our careful response together with additional experiments could help mitigate the concerns.
>
> ### W1- How IGC is maintained:
> **Answer:**
> In general, we preserve IGC by explicitly disentangling exploration and mode. Please refer to Common Response CR1.
>
> ### W2- Incomplete results + limited ablation studies:
> **Answer:**
> The curve of AengMixer is obscured by the curve of MAT. We changed the color to make Figure 3 clearer. We conducted ablation study demonstrating the effect of full state information and comparing a single-agent RL algorithm on the joint action space (AIL). Please refer to Common Response CR3.
>
> ### W3- Weak improvement:
> **Answer:**
> The main focus of this work is addressing asymmetric information failure. The performance of AgentMixer is highly related to how severe the partial observation of the task is. Please also refer to Common Response CR2&3.
>
> ### W4- Missing references:
> **Answer:**
> We cited the suggested reference. To handle partial observability, Centralized Training with Decentralized Execution (CTDE) is proposed as a popular learning framework for MARL. In existing CTDE methods, including value decomposition and policy gradient methods, decentralized agent policies are trained by a centralized (action-)value function with additional global information, while agents make decisions only based on their own local observation. However, the centralized training in CTDE is not centralized enough as agent policies are assumed to be independent of each other. Besides, the CTDE framework only introduces global information in the value function while policies are not granted direct access to global information even when centralized training. As far as we know, AgentMixer is the first work introducing both global information and dependencies into policy gradient.
>
> ### W5- Improve notation:
> We apologize for the unclear notations and typos.
> **Answer:**
> > difference in notation or wording between Definition 1 and 2
> >
> The difference between Definition 1 and 2 is that 1 is for product policy while 2 is for joint policy.
>
> > confusing notation
> >
> $w_{agent}$ and $w_{channel}$ denote different fully connected layers used in agent- and channel-mixer. $f$ denotes a strategy modification defined in Section 3.2.
>
> > location of Figure 2
>
> We modified Figure 2 and its location.
>
> > Typos
> >
> We fixed the typos.
>
> > placement of the related works
>
> We rearranged the placement of the related works.
>
> > Confusion about Definition 4
>
> We added a discussion of this posterior inference procedure in Section 4.2. Implicit product policy is defined as a posterior inference procedure, marginalizing the conditional occupancy $\rho^{\pi_{\psi }}(s|b)$. Since the observations/belief may not contain information to distinguish two different latent states, the $\rho^{\pi_{\psi }}(s|b)$ is a stochastic distribution, and the implicit product policy is the average of the fully observable policy.
>
> ### W1- Why AgentMixer performs best in Ant?
> **Answer:**
> The reason why AgentMixer achieves superior performance on Ant-v2 where partial observability poses a critical challenge is that AgentMixer keeps consistency between joint fully observable policy and individual partially observable policies. To verify this, we presented ablation studies. Please also refer to Common Response 2&3.
>
> ### W2- Additional experiment on Predator-Prey:
> **Answer:**
> We conducted experiments on the predator-prey w/punishment domain in Appendix. Figure 12 shows that AgentMixer can effectively alleviate the relative overgeneralization problem.

---

> > ### Author Response · Authors · 2023-11-21
> > **A Gentle Reminder**
> >
> > Dear Reviewer tDVs:
> >
> > Thank you for your time and dedication again in reviewing our paper. In our rebuttal, we have provided a more comprehensive explanation and additional experiments for the proposed framework. We believe that we have thoroughly and effectively addressed your comments.
> >
> > We kindly remind you that the discussion period will end soon (in a few days). Should you have any additional questions or require further clarification, please feel free to reach out. We appreciate your time and consideration and we would appreciate it if you could revise your scores based on the changes made.
> >
> > Best regards,
> >
> > Authors

---

> > > ### Comment · Reviewer_tDVs · 2023-11-21
> > > **Updated score**
> > >
> > > Thanks to the authors for their revisions. The newly added sections on the implementation of AgentMixer in the discrete/continuous cases improve my understanding of the method. I also appreciate the addition of AIL as a baseline. As such, I have raised my score.
> > >
> > >
> > > However, I still don't think the paper is ready for acceptance. I'm still not sure why AgentMixer does better on Ant-v2. The authors claim it is because partial observabalitity is a bigger challenge on Ant-v2 than other domains, but what evidence do they have?
> > >
> > > Second, it still needs a lot of writing improvement -- there are too many ambiguities, which makes reading and understanding very challenging. I found this to be the case even in the newly added sections. Finally, seeing as partial observability doesn't seem to greatly impact performance in Starcraft v2, I encourage the authors to explore alternative domains which might better showcase the benefits of their method.

---

> > > > ### Author Response · Authors · 2023-11-22
> > > > **Response to Reviewer tDVs**
> > > >
> > > > We thank the reviewer for their feedback and engagement in the discussion process. It is great to hear that the reviewer found our responses helpful.
> > > >
> > > > >  I'm still not sure why AgentMixer does better on Ant-v2. The authors claim it is because partial observabalitity is a bigger challenge on Ant-v2 than other domains, but what evidence do they have?
> > > >
> > > > **Answer:**
> > > > Compared to HalfCheetah-v2, the local observations of each agent (leg) of ants are very similar, making it difficult to estimate the state information required for coordination. To verify this, we added an ablation experiment compared with MAPPO with full observation during execution, MAPPO_FULL, in Figure 10. We can see that MAPPO_FULL with full observation only achieves a similar performance with our method. This result confirms our conclusion.
> > > >
> > > > > Second, it still needs a lot of writing improvement -- there are too many ambiguities, which makes reading and understanding very challenging.
> > > >
> > > > **Answer:**
> > > > We apologize for the unclear statements. We will carefully proofread the writing and submit revisions as soon as possible.
> > > >
> > > > > Finally, seeing as partial observability doesn't seem to greatly impact performance in Starcraft v2, I encourage the authors to explore alternative domains which might better showcase the benefits of their method.
> > > >
> > > > **Answer:**
> > > > We are very grateful for your constructive suggestion. As we are approaching the end of the author-reviewer discussion, it may be difficult for us to find a suitable domain in a short time. In addition to this, we conducted experiments on the predator-prey w/punishment domain according to your suggestion. Results show that AgentMixer successfully learns fully coordinated decentralized policies.

---

> ### Author Response · Authors · 2023-11-23
> **A kind reminder**
>
> Dear Reviewer tDVs,
>
> We would like to express our sincere gratitude to you for reviewing our paper and providing valuable feedback. We believe that we have responded to and addressed all your concerns with our response and revision.
>
> Notably, given that we are approaching the deadline for the rebuttal phase, we hope we can have the discussion soon. Thanks！
>
> Best,
>
> All authors

---

### Official Review · Reviewer_Qpfs · 2023-10-31

**Soundness:** 2 fair
**Presentation:** 2 fair
**Contribution:** 2 fair
**Rating:** 5
**Confidence:** 4

**Summary:**

In this paper, the authors target the challenge of stabilizing partially observable multi-agent reinforcement learning (MARL) by proposing AgentMixer, a novel approach that leverages centralized training with decentralized execution (CTDE) to learn correlated decentralized policies. AgentMixer consists of a Policy Modifier (PM) module that models the correlated joint policy and Individual-Global-Consistency (IGC) that maintains consistency between individual policies and joint policy while allowing correlated exploration. They theoretically prove that AgentMixer converges to ϵ-approximate Correlated Equilibrium. Furthermore, they evaluate AgentMixer on three MARL benchmarks, demonstrating its effectiveness in handling partial observability and achieving strong experimental performance.

**Strengths:**

- Originality: The paper introduces AgentMixer, a novel MARL approach that combines Policy Modifier and Individual-Global-Consistency to address partial observability challenges.
- Quality: The theoretical analysis provided includes convergence to an ϵ-approximate Correlated Equilibrium, which showcases the robustness of the proposed approach.
- Significance: Experimental performance shows that AgentMixer outperforms existing state-of-the-art MARL methods on three benchmarks, confirming the method's effectiveness and applicability to real-world problems.

**Weaknesses:**

1. Some claims about existing works may be inappropriate;
2. The method part is not clear written;
3. The experiments can be improved with more critical baselines and better aligning the motivation.

**Questions:**

1.  "While this is attractive, the pre-defined dependence among agents in auto-regressive methods may limit the representation expressiveness of their joint policy classes. " Why auto-regressive limit the expressiveness? [1]'s eq (5 ) and [2]'s Theorem A.1 provide opposition for the claim.
2. The claim about " However, note that both auto-regressive methods and existing correlated policies violate the requirement for decentralized execution. "  is questionable. As auto-regreesive[1]  and correlated[3] are decentralized execution.
3. In the method part, eq6, what is the superscript of W_agent mean? why we specially need channel mixing? can you provide some intuitions?
4. For definition 5, why should we care about identifiability? as in 3, we only need the divergence between local one and global optimal as closer as it can be. Does the ``closer'' surely be the mode consistent?
5. For Figure 2, you not show show IGC. Also hwo your IGC been used? as a constraint on optimization of 13? Then can you concretely write it and derive the grad?
6. [1] is very relevant and I do think it should be involved in your baseline.


- Minors
7. The [1], [2], and [3] should be cited.
8. The IGC should be clearly showed in both loss function and your main figure.
9. The SMAC-v2 seems still not a good testbed for your motivation or method, I suggest you put it to appendix.

---
[1] Wang, Jiangxing, Deheng Ye, and Zongqing Lu. "More Centralized Training, Still Decentralized Execution: Multi-Agent Conditional Policy Factorization." ICLR 2022.

[2] Sheng, Junjie, et al. "Negotiated Reasoning: On Provably Addressing Relative Over-Generalization." arXiv preprint arXiv:2306.05353 (2023).

[3] Wen, Ying, et al. "Probabilistic recursive reasoning for multi-agent reinforcement learning." ICLR 2019.

---

> ### Author Response · Authors · 2023-11-17
> **Response to Reviewer Qpfs**
>
> ## Response to Reviewer Qpfs:
>
> We appreciate your constructive comments and apologize for the unclear statements in our paper. We believe our careful response and additional experiments could help mitigate your concerns.
>
> ### W1 and Q1,2- Inappropriate claims on auto-regressive methods:
> **Answer:**
> We thank you for this constructive suggestion. We have modified the corresponding texts and cited the suggested reference. However, note that existing auto-regressive methods assume a pre-defined execution order, while our method doesn't assume any execution order.
>
> ### W2 and Q5- How IGC is used:
> **Answer:**
> We apologize for the unclear explanation. In general, we preserve IGC by explicitly disentangling exploration and mode. We believe Common Response CR1 has clarified this question.
>
> ### Q3- Notation of $w_{agent}$ and why channel mixing:
> **Answer:**
> $w_{agent}$ and $w_{channel}$ denote different fully connected layers used in agent- and channel-mixer. The agent-mixer allows 'communication' between agents' dims, which operates on each dim/channel independently (each agent has $m$ feature dims). The channel-mixer operates on each agent independently allowing 'communication' between different channels. These two types of mixers are interleaved to enable interaction of both input dimensions (agents and dims).
>
> ### Q4- Why care about identifiability?
> >For definition 5, why should we care about identifiability? as in 3, we only need the divergence between local one and global optimal as closer as it can be. Does the ``closer'' surely be the mode consistent?
>
> **Answer:**
> Identifiability is the crucial insight to guarantee that the partially observed policy recovered through imitation learning can exactly reproduce the actions of the fully observing policy. According to the definition of identifiability, identifiability is measured by the divergence between joint policy and individual policies. However, directly adding divergence constraint to learning of individual policies may lead to unexpected results. To verify this, we compared with asymmetric imitation learning (AIL) where individual policies directly imitate from a centralized PPO with full state information. Please also refer to Common Response CR3.
>
> ### Q6- Additional baseline of MACPF:
> **Answer:**
> We added MACPF as one of the baselines.

---

> > ### Author Response · Authors · 2023-11-21
> > **A Gentle Reminder**
> >
> > Dear Reviewer Qpfs:
> >
> > Thank you for your time and dedication again in reviewing our paper. In our rebuttal, we have provided a more comprehensive explanation and additional experiments for the proposed framework. We believe that we have thoroughly and effectively addressed your comments.
> >
> > We kindly remind you that the discussion period will end soon (in a few days). Should you have any additional questions or require further clarification, please feel free to reach out. We appreciate your time and consideration and we would appreciate it if you could revise your scores based on the changes made.
> >
> > Best regards,
> >
> > Authors

---

> > > ### Comment · Reviewer_Qpfs · 2023-11-22
> > > **Thank you for your thorough and detailed response**
> > >
> > > Thank you for your thorough and detailed response addressing most of my concerns and questions. I appreciate the effort that has gone into conducting additional experiments during the rebuttal period, which is no easy task, and I recognize the hard work that you have put forth.
> > >
> > > However, I still believe that the most critical concern has not been adequately addressed. The major motivation of this paper is the insufficient representational capacity of autoregressive methods. As I mentioned in my previous communication, some existing approaches have presented contrary conclusions. Although you have clarified in the revised version that autoregressive methods require a fixed execution order for agent actions, a deeper analysis comparing the representational capacity of autoregressive methods with AgentMixer is lacking.
> > >
> > > Similarly, while the experimental section now includes MACPF as one of the baselines and demonstrates superior performance by AgentMixer, I feel that a mere numerical comparison is insufficient.
> > >
> > > In summary, considering the current version's inadequacy in discussing the representational capacity of autoregressive methods and its comparison to AgentMixer, I believe it has not yet reached the standard for publication. Therefore, I maintain my original score for the paper.

---

> > > > ### Author Response · Authors · 2023-11-22
> > > > **Response to Reviewer Qpfs**
> > > >
> > > > We sincerely appreciate the reviewer’s timely response and further discussions. Below, we try our best to address your new concerns.
> > > >
> > > > >  The major motivation of this paper is the insufficient representational capacity of autoregressive methods. As I mentioned in my previous communication, some existing approaches have presented contrary conclusions. Although you have clarified in the revised version that autoregressive methods require a fixed execution order for agent actions, a deeper analysis comparing the representational capacity of autoregressive methods with AgentMixer is lacking.
> > > >
> > > > **Answer:**
> > > > We apologize for the misleading terminology. However, we would like to emphasize that the motivation of this work is to alleviate the ***asymmetric information failure*** in CTDE caused by the mismatch between global information and local information. As mentioned in Common Response, AIL shows ***severe asymmetric performance*** during training and testing phases, which confirms our motivation. In addition, we theoretically confirmed the cause of this failure. Therefore, our method does not conflict with the autoregressive methods, and ***does not*** aim at improving representational capacity.

---

> ### Comment · Reviewer_Qpfs · 2023-11-22
> **Thanks for your response**
>
> I would like to thank you for your response. I understand that the core of this paper is emphasizing the importance of asymmetric learning failure and proposing AgentMixer to address this issue. The point I wanted to emphasize in my previous response was that certain autoregressive methods (such as those mentioned in Question 2 [1,3]) **can also achieve decentralized execution**; however, the paper does not explicitly analyze why these methods **cannot** solve the asymmetric learning failure problem.
>
> Furthermore, regarding the proof-of-concept task used in section 5.1 of the paper, its purpose should be to answer the aforementioned question from an experimental perspective, i.e., why AgentMixer can solve the asymmetric learning failure problem, but the baselines cannot. However, I cannot intuitively establish a connection between this task and the asymmetric learning failure problem, and the paper introduces the new issue of relative overgeneralization. The paper does not provide a clear explanation of the relationship between this issue and the asymmetric learning failure problem.
>
> Additionally, the authors mentioned an interesting question during their discussion with Reviewer 9QDT, namely the **relationship between IGC and IGM**. The authors suggested that, **informally**, IGC is similar to an IGM for policies. Given the relationship between optimal policies and optimal Q value functions from the probabilistic graphical model perspective [^], can IGC be easily extended from IGM based on this relationship? I believe a discussion of this aspect could more clearly position this paper.
>
> In summary, I believe that the asymmetric learning failure problem addressed in this paper, as well as the proposed IGC concept and AgentMixer algorithm, contribute to the MARL community. However, in the current version, the paper's content does not present a logical flow throughout, and the experiments (particularly the proof-of-concept task) do not fully corroborate the series of conclusions presented in the paper. If the authors can better address these issues, I will raise my score to 6.
>
> [^] Levine, Sergey. "Reinforcement learning and control as probabilistic inference: Tutorial and review." arXiv preprint arXiv:1805.00909 (2018).

---

> > ### Author Response · Authors · 2023-11-22
> > **Response to Reviewer Qpfs**
> >
> > We thank the reviewer Qpfs for engaging in the discussion and providing insightful feedback! Below, we try our best to address your new concerns.
> >
> > >  Analyze why these methods cannot solve the asymmetric learning failure problem
> >
> > **Answer**
> > PR2^[1], a self-play-based recursive reasoning algorithm that follows the decentralized-training-decentralized-execution paradigm, aimed to solve the decision-making problem in Multi-Agent MDPs, and there was no partially observable problem. For MACPF^[2], it introduces auto-regressive model to FOP^[3] and separates the decentralized policies from the centralized policy. However, under partial observability, since the dependency policy correction term conditioned on global information contributes more to the dependent policy while the independent policy can only learn the average of the dependent policy under conditional occupancy (Definition 4), it may still suffer from asymmetric learning failure.
> >
> > > The paper does not provide a clear explanation of the relationship between the relative overgeneralization issue and the asymmetric learning failure problem.
> >
> > **Answer**
> > We apologize for the unclear statements. The relative overgeneralization issue is unrelated to the asymmetric learning failure problem. The purpose of the experiment in the matrix game is to verify whether PM can model correlated joint policy and whether IGC can guarantee the consistency of joint policy and individual policies.
> >
> > > Relationship between IGC and IGM
> >
> > **Answer**
> > Thanks for your constructive suggestion! We believe that IGC and IGM are parallel works on policy gradient- and value-based methods respectively, and can be equivalently transformed. Returning to the previous concern, another reason why MACPF suffers from asymmetric learning failure is that the IGM between dependent critic and independent critic is not satisfied.
> >
> > [1] Wen, Ying, et al. "Probabilistic recursive reasoning for multi-agent reinforcement learning." ICLR 2019.
> > [2] Wang, Jiangxing, Deheng Ye, and Zongqing Lu. "More Centralized Training, Still Decentralized Execution: Multi-Agent Conditional Policy Factorization." ICLR 2022.
> > [3] Tianhao Zhang, Yueheng Li, Chen Wang, Guangming Xie and Zongqing Lu. "Fop: Factorizing optimal joint policy of maximum-entropy multi-agent reinforcement learning." ICML 2021.

---

> > > ### Comment · Reviewer_Qpfs · 2023-11-23
> > > **Update the score**
> > >
> > > Thank you for the prompt response. Your discussion above has helped address my main concerns and provided me with a deeper understanding of the contributions and positioning of this work in the context of MARL. As a result, I have raised my score to a 5. Nevertheless, I still believe that additional revision round is necessary to address these issues and ensure that the paper meets publication standards.

---

> > > > ### Author Response · Authors · 2023-11-23
> > > > **Response to Reviewer Qpfs**
> > > >
> > > > Thank you so much again for your valuable suggestions and for raising your score! We now further address your concerns.
> > > >
> > > > >  The point I wanted to emphasize in my previous response was that certain autoregressive methods (such as those mentioned in Question 2 [1,3]) can also achieve decentralized execution; however, the paper does not explicitly analyze why these methods cannot solve the asymmetric learning failure problem.
> > > >
> > > > **Answer**
> > > > For MACPF, the lack of restrictions on dependent and independent policies may lead to inconsistencies. Considering the scenario where the agents learn to cross the bridge in Figure 1. The independent policies conditioned on local information tend to learn a deterministic policy, selecting path 1. In contrast, the dependency policy correction term conditions on global information, allowing it to learn a conditional policy contingent upon the physiques of the other agent. Specifically, the correction term introduces none or rightward deviations from path 1. Hence, independent policies and dependent policies yield divergent decisions, giving rise to the asymmetric learning failure problem.
> > > >
> > > > > Furthermore, regarding the proof-of-concept task used in section 5.1 of the paper, its purpose should be to answer the aforementioned question from an experimental perspective, i.e., why AgentMixer can solve the asymmetric learning failure problem, but the baselines cannot. However, I cannot intuitively establish a connection between this task and the asymmetric learning failure problem, and the paper introduces the new issue of relative overgeneralization. The paper does not provide a clear explanation of the relationship between this issue and the asymmetric learning failure problem.
> > > >
> > > > **Answer**
> > > > This work does not aim to address the relative overgeneralization problem, which causes the agents to get stuck into local optima. We use the matrix game as an illustrative example to demonstrate the effectiveness of PM and IGC. The baselines failed in this domain as they assumed independence among agents. We have modified the corresponding analysis for this experiment.
> > > >
> > > > > Given the relationship between optimal policies and optimal Q value functions from the probabilistic graphical model perspective [^], can IGC be easily extended from IGM based on this relationship?
> > > >
> > > > **Answer**
> > > > We appreciate your perspective on the IGC and IGM relationship from this perspective! IGC may potentially emerge from the Boltzmann machine representation of IGM.
> > > >
> > > > Should you believe our revisions have successfully addressed the issues raised, we kindly request you to adjust the scores accordingly. Your support is greatly appreciated!

---

### Official Review · Reviewer_9QDT · 2023-11-02

**Soundness:** 3 good
**Presentation:** 3 good
**Contribution:** 2 fair
**Rating:** 5
**Confidence:** 3

**Summary:**

This paper discusses the issue that agents make decisions based on their local observation independently, which could hardly lead to a correlated joint policy with sufficient coordination.

Two key ideas are introduced, i.e., Policy Modifier and Individual-Global-Consistency. Policy Modifier takes the individual partially observable policies and state as inputs and produces correlated joint fully observable policy as outputs. Individual-Global-Consistency keeps the mode consistency among the joint policy and individual policies.

Overall, the presentation is clear, while the authors have clearly expressed their results. The authors have clearly explained the MARL problem they wanted to solve and how to solvem. However, the necessity and importance of work have not been clearly expressed, and there are still doubts which are explained in details as follows.

**Strengths:**

The core of multi-agent reinforcement learning is strategy alignment and we need to align the local policies obtained from local observation information with the joint policy. This paper aims to solve this issue by polishing the obtained local policies following a well-defined Individual-Global-Consistency condition.

**Weaknesses:**

The given Individual-Global-Consistency condition seems to be too high-level, which is similar with Individual-Global-Max condition for value decomposition MARL methods. The authors directly consider this condition as the constraint in (13). The most important issue of the proposed algorithm is that we need to check whether this constraint can be satisfied by following the proposed algorithm. If not, can we measure this distance? Both theoretically and experimentally? This has impaired the contribution of this work.

**Questions:**

1. Clarify the connection between IGC and IGM, while compare the corresponding algorithms would be better.
2. Indeed, there are limitations to the length of the paper, and it is still recommended that the author provide a clear algorithm pesudo code.
3. The authors discussed the performance of the algorithm during the experimental phase, but did not validate the key technologies proposed in this article, especially the issue of whether modifying local polies can lead to better alignment of joint policy.
4. I still emphasize the necessity of emphasizing CTDE. This article should be compared with value decomposition or strategy decomposition methods, rather than emphasizing CTDE as a computational framework.

**Details Of Ethics Concerns:**

~

---

> ### Author Response · Authors · 2023-11-17
> **Response to Reviewer 9QDT**
>
> ## Response to Reviewer 9QDT:
> We thank you for your kind comments and apologize for the unclear statements in our paper. We believe our careful response together with additional experiments could help mitigate the concerns.
>
> ### W1- How IGC is satisfied:
>
> > The most important issue of the proposed algorithm is that we need to check whether this constraint can be satisfied by following the proposed algorithm. If not, can we measure this distance?
>
> **Answer:**
> In general, we preserve IGC by explicitly disentangling exploration and mode. Please also refer to Common Response CR1 where we provide a detailed explanation.
>
> ### Q1- Connection between IGC and IGM:
> > Clarify the connection between IGC and IGM, while compare the corresponding algorithms would be better.
> >
> **Answer:**
> Thanks for your constructive insights! We find that both IGM and IGC aim to maintain the optimal action consistent between joint policy and individual policies. While IGM is imposed on joint action-value function (Q value) and individual Q value, IGC **explicitly maintains mode consistency** between joint policy and individual policies. We believe such a connection may benefit the MARL community more.
>
> One of the compared baselines, MAVEN, based on QMIX requires IGM. As far as we know, AgentMixer is the first work that imposes IGC on policies.
>
> ### Q2- Pseudo code:
> **Answer:**
> Thanks for your constructive suggestion. We added a pseudo code in Appendix.
>
> ### Q3- Validate the key technologies:
> > The authors discussed the performance of the algorithm during the experimental phase, but did not validate the key technologies proposed in this article, especially the issue of whether modifying local polies can lead to better alignment of joint policy.
>
> **Answer:**
> To validate AgentMixer, we perform ablation experiments by adding two baselines, MAPPO_FULL conditioned on global information in both training and testing; asymmetric imitation learning (AIL) which uses fully observable PPO to supervise learning decentralized partially observable policies. Results demonstrate that AgentMixer mitigates asymmetric information failure by guaranteeing consistency between joint policy and individual policies. Please also refer to Common Response CR1,2,3.
>
> ### Q4- Compare with value decomposition or strategy decomposition methods:
> > I still emphasize the necessity of emphasizing CTDE. This article should be compared with value decomposition or strategy decomposition methods, rather than emphasizing CTDE as a computational framework.
> >
> **Answer:**
> The baselines compared are all based on CTDE, except for MAT and MAT-dec which are CTCE (centralized training with centralized execution). MAVEN and MACPF are value decomposition and policy decomposition methods respectively.

---

> > ### Author Response · Authors · 2023-11-21
> > **A Gentle Reminder**
> >
> > Dear Reviewer 9QDT:
> >
> > Thank you for your time and dedication again in reviewing our paper. In our rebuttal, we have provided a more comprehensive explanation and additional experiments for the proposed framework. We believe that we have thoroughly and effectively addressed your comments.
> >
> > We kindly remind you that the discussion period will end soon (in a few days). Should you have any additional questions or require further clarification, please feel free to reach out. We appreciate your time and consideration and we would appreciate it if you could revise your scores based on the changes made.
> >
> > Best regards,
> >
> > Authors

---

> ### Author Response · Authors · 2023-11-22
> **A kind reminder**
>
> Dear Reviewer 9QDT,
>
> We would like to express our sincere gratitude to you for reviewing our paper and providing valuable feedback. We believe that we have responded to and addressed all your concerns with our response and revision.
>
> Notably, given that we are approaching the deadline for the rebuttal phase, we hope we can have the discussion soon. Thanks！
>
> Best,
> All authors

---

### Official Review · Reviewer_yFNn · 2023-11-06

**Soundness:** 3 good
**Presentation:** 2 fair
**Contribution:** 3 good
**Rating:** 3
**Confidence:** 3

**Summary:**

This manuscript aims to the asymmetric learning failure problem in Centralized training with decentralized execution (CTDE). To fully take advantage of CTDE to learn correlated decentralized policies, the authors propose the AgentMixer algorithm. It has two key module. The first one is Policy Modifier (PM), which explicitly models the correlated joint policy via composing the partially observable individual policies conditioned on global state information. The second one is Individual-Global-Consistency (IGC), which maintains the mode consistent between the state-based joint policy and partially observable decentralized policies. It is theoretically proofed that AgentMixer converges to ε-approximate Correlated Equilibrium, and the experimental results show that it can achieve comparable performance to existing methods.

**Strengths:**

This paper present a novel solution to factorize the joint policy in MARL, i.e. policy mixing and distilling. It also introduce a ε-approximate Correlated Equilibrium perspective to measure the consistence and propose the Individual-Global-Consistency (IGC) to guarantee.

**Weaknesses:**

Although it's theoretically proofed that AgentMixer converges to ε-approximate Correlated Equilibrium, the results in the experiments show that the performance of AgentMixer is even or not better than compared methods in many settings.
The convergence of AgentMixer is also proofed via the mode consistency. But, The IGC is defined based on the mode of the policy distribution, and the PM is defined via MLP mixer (agent- and channel). It may have some gap here.
Besides, some contents seem inconsistent in the presentation, figures in the manuscript.

**Questions:**

1. Is the Figure 2 a final version? There are no Individual-Global-Consistency components, the gradient forward and backward procedure is not mentioned in the content, and where is the policy distilling?
2. The performance of AgentMixer is shown even or not better than compared methods in many experiments. The authors should give more analysis.
3. It may need to be more clear, that how the policy modifier (mixing) can keep the mode consistency.
4. Some details are not given clearly in the manuscripts, ie.
    How does the embedding is achieved with $\pi_{\theta_1}$ and $s$?
    what is the meaning of $b$ in the product policy $\pi_{\phi}(a|b)$?
5. The information of this paper cited is incomplete.
Jianing Ye, Chenghao Li, Jianhao Wang, and Chongjie Zhang. Towards global optimality in cooperative marl with the transformation and distillation framework, 2023.

**Details Of Ethics Concerns:**

None.

---

> ### Author Response · Authors · 2023-11-17
> **Response to Reviewer yFNn**
>
> ## Response to Reviewer yFNn:
> We thank you for your constructive comments. We have done additional experiments and explaination and hope it can address your concerns.
>
> ### W1, Q2- Flat performance:
> > the results in the experiments show that the performance of AgentMixer is even or not better than compared methods in many settings
>
> **Answer:**
> The main focus of this work is addressing asymmetric information failure. The performance of AgentMixer is highly related to how severe the partial observation of the task is. To justify this, we perform ablation experiments by adding two baselines, MAPPO_FULL conditioned on global information in both training and testing; asymmetric imitation learning (AIL) which uses fully observable PPO to supervise learning decentralized partially observable policies. As shown in Figure 6, directly distilling from joint policy will suffer from asymmetric information failure while our method achieves strong performance by mitigating asymmetric information failure. Results in Figure 12 demonstrate that when global information is not critical, the compared methods perform similarly. Please also refer to Common Response CR2&3.
>
> ### W2- Gap between IGC and PM:
> > The IGC is defined based on the mode of the policy distribution, and the PM is defined via MLP mixer (agent- and channel). It may have some gap here.
>
> **Answer:**
> PM takes as input the individual partially observable policies and state and outputs correlated joint fully observable policy, where the agent- and channel-mixer aim to introduce dependency among agents. IGC enables the actions that occur most frequently in the joint policy and the individual policies to be equivalent, which keeps consistency between the joint policy and individual policies, thereby mitigating asymmetric information failure. **In short, PM achieves IGC by only modifying the exploration of individual policies, for example, the std. in Gaussian policy, while maintaining the mean (mode) unchanged.** Please also refer to Common Response CR1 for a detailed explanation.
>
> ### Q1- Unclear Figure 2:
> > There are no Individual-Global-Consistency components, the gradient forward and backward procedure is not mentioned in the content, and where is the policy distilling?
>
> **Answer:**
> We modified Figure 2 to clarify the relationship among components in AgentMixer.
>
> ### Q3- Mode consistency:
> > It may need to be more clear, that how the policy modifier (mixing) can keep the mode consistency.
> >
> **Answer:**
> IGC aims to keep the mode of joint policy and individual policies consistent. In general, we preserve IGC by explicitly disentangling exploration and mode. Please also refer to Common Response CR1.
>
> ### Q4- Notation details:
> > Some details are not given clearly in the manuscripts
>
> **Answer**
> We added some notation for clarity. The embedding of $\pi_{\phi}$ and $s$ is achieved by MLP. $b$ is a belief state which is a sufficient statistic for joint history.
>
> ### Q5- Suggested citation:
> > The information of this paper cited is incomplete. Jianing Ye, Chenghao Li, Jianhao Wang, and Chongjie Zhang. Towards global optimality in cooperative marl with the transformation and distillation framework, 2023.
>
> **Answer**
> We modified this citation.

---

> > ### Author Response · Authors · 2023-11-21
> > **A Gentle Reminder**
> >
> > Dear Reviewer yFNn:
> >
> > Thank you for your time and dedication again in reviewing our paper. In our rebuttal, we have provided a more comprehensive explanation and additional experiments for the proposed framework. We believe that we have thoroughly and effectively addressed your comments.
> >
> > We kindly remind you that the discussion period will end soon (in a few days). Should you have any additional questions or require further clarification, please feel free to reach out. We appreciate your time and consideration and we would appreciate it if you could revise your scores based on the changes made.
> >
> > Best regards,
> >
> > Authors

---

> ### Author Response · Authors · 2023-11-22
> **A kind reminder**
>
> Dear Reviewer yFNn,
>
> We would like to express our sincere gratitude to you for reviewing our paper and providing valuable feedback. We believe that we have responded to and addressed all your concerns with our response and revision.
>
> Notably, given that we are approaching the deadline for the rebuttal phase, we hope we can have the discussion soon. Thanks！
>
> Best,
> All authors

---

### Author Response · Authors · 2023-11-17
**Common Response**

## Common Response with Important Clarifications:
We thank every reviewer for your valuable time in reviewing our paper. We summarize the main concerns shared by reviewers and hope our response can clarify them.

### CR1. How to implement IGC:
Thank you for raising this concern. In general, we preserve IGC by explicitly disentangling exploration and mode. We divide the implementation of IGC into two categories (in Section 4.2.1 and 4.2.2): continuous action space and discrete action space.
For continuous action space, we assume the policy as a Gaussian distribution and **keep the mean of joint policy the same as individual policies while the standard deviation is generated by PM**.
For discrete action space, we assume individual policies as Categorical distributions and joint policy as Gumbel-Softmax distribution where **logits are the same as individual policies while the Gumbel noise is generated by PM**.

### CR2. Flat performance in SMAC-v2 tasks:
In this work, we mainly focus on mitigating asymmetric information failure which is caused by the mismatch between global information and local information. The superior performance of AgentMixer on Ant-v2 where partial observability poses a critical challenge confirms our motivation. The reason why AgentMixer achieves flat performance compared with the baselines in SMAC-v2 tasks is that global information makes a relatively small impact on learning in such a domain. To verify this, we conduct an ablation study demonstrating the effect of full state information on SMAC-v2. Results in Figure 11 show that **even centralized execution with full state information achieves flat performance**.

### CR3. Added Ablation study:

In order to empirically show the *asymmetric learning failure* problem, we construct additional baselines:

- **Asymmetric imitation learning (AIL)** which directly distills partial observation policies from a centralized PPO with full state information.
- MAPPO with full observation during execution, **MAPPO_FULL**. This method can be seen as the oracle baseline with advantageous information.

Additional experiments in Figure 6 show that AIL enjoys significant training performance as the interaction with the environment is based on full information. However, **the distilled decentralized policies fail to learn any meaningful policies**. This conflicting result validates the *asymmetric learning failure* we aim to solve.

Note that **MAPPO_FULL with full observation only achieves similar performance** with our method, as well as some baselines in the SMAC_v2 domain, as shown in Figure 11 in the Appendix. We can conclude that **the difficulty of SMAC_v2 doesn't come from the partial observation that our method aims to mitigate**. It thus helps explain our comparable results with the baselines in Figure 5 and 11. However, our method outperforms the baselines when considering the whole extensive experiments.

### CR4. Summary of Different CTDE settings of Baselines:
We summarize the different CTDE settings used in baselines in the following table. We can find that although all the methods take advantage of CTDE by learning a centralized value (Cent. Value) during training, **only AIL and our method further employ a centralized joint policy (Cent. Policy)**. Note that MAT and MAT-Dec access the full state information during execution, which is an unfair comparison in evaluation.

| Algorithm               | MAPPO   | HAPPO   | MAT     | MAT-Dec | MAVEN   | MACPF   | AIL     | Ours    |
|-------------------------|---------|---------|---------|---------|---------|---------|---------|---------|
| Partial obs.(execution) | &check; | &check; | &cross; | &cross; | &check; | &check; | &check; | &check; |
| Cent. Value (training)  | &check; | &check; | &check; | &check; | &check; | &check; | &check; | &check; |
| Cent. Policy (training) | &cross; | &cross; | &cross; | &cross; | &cross; | &cross; | &check; | &check; |

---

### Author Response · Authors · 2023-11-20
**General response to the reviewers**

Dear Reviewers,

Thanks again for your valuable comments and suggestions. We are pleased that the reviewers appreciated the **novelty** of AgentMixer (yFNn, Qpfs), recognized that the problem is **clearly stated** (9QDT), and found the idea of correlated policy factorization **interesting** (tDVs). We notice that the main concerns of the reviewers come from the lack of explanation of IGC and the performance of AgentMixer. During the rebuttal period, we conducted extensive ablation studies and enhanced the overall completeness. These efforts aimed to enhance readability and clearly articulate the novelty of our work. We have submitted our rebuttals to your reviews. We explain the problems and try our best to settle the reviewers' concerns, including paper revisions and additional experimental results.
We look forward to your response and are eager to continue our discussion.

Sincerely,

Authors

---

### Author Response · Authors · 2023-11-22
**Main Contribution**

Thanks to all reviewers for the constructive comments. We highlight here the contribution of this work.

One of the **core problems** that we try to address here is the ***mismatch between global information and local information***, which is commonly ignored in CTDE. There is a notable absence of research on ***asymmetric learning failure*** within the MARL context, let alone the formulation of theoretical frameworks in this domain. To the best of our knowledge, AgentMixer is the **first** to consider the issue in MARL.

To verify this phenomenon, we constructed additional baselines, AIL and MAPPO_FULL. The results confirm the asymmetric learning failure we aimed to address. To address this challenge, we draw upon some of the proof techniques from papers like Warrington et al. (2021). We utilize certain existing lemmas and theorems, on top of which we provide novel theoretical formulations, mechanism (IGC), and new results as seen in Theorem 1 and Theorem 2, and their accompanied proofs in Appendix which are our primary contributions.

---

### Author Response · Authors · 2023-11-22
**Looking forward to your reply before deadline**

Dear Reviewers:

Thanks again for your constructive suggestions. We have tried our best to address your concerns. For instance, we have reorganized our paper to improve the presentation and also provided more extensive experimental results in not only the main text but also the appendix. We believe that these modifications improve our manuscript significantly. As the deadline for rolling discussion (Nov 22nd) approaches, we would be happy to take this opportunity to have more discussions. If our rebuttal has addressed your concerns, we would be grateful if you could re-evaluate our paper.

Thanks!

Best,

Authors